# RPAP3 provides a flexible scaffold for coupling HSP90 to the human R2TP co-chaperone complex

Fabrizio Martino[1], Mohinder Pal[2], Hugo Muñoz-Hernández[1,3], Carlos F. Rodríguez[1,3], Rafael Núñez-Ramírez[1], David Gil-Carton[4], Gianluca Degliesposti[5], J. Mark Skehel[5], S. Mark Roe[2], Chrisostomos Prodromou[2], Laurence H. Pearl [2] & Oscar Llorca [1,3]

The R2TP/Prefoldin-like co-chaperone, in concert with HSP90, facilitates assembly and cellular stability of RNA polymerase II, and complexes of PI3-kinase-like kinases such as mTOR. However, the mechanism by which this occurs is poorly understood. Here we use cryo-EM and biochemical studies on the human R2TP core (RUVBL1–RUVBL2–RPAP3–PIH1D1) which reveal the distinctive role of RPAP3, distinguishing metazoan R2TP from the smaller yeast equivalent. RPAP3 spans both faces of a single RUVBL ring, providing an extended scaffold that recruits clients and provides a flexible tether for HSP90. A 3.6 Å cryo-EM structure reveals direct interaction of a C-terminal domain of RPAP3 and the ATPase domain of RUVBL2, necessary for human R2TP assembly but absent from yeast. The mobile TPR domains of RPAP3 map to the opposite face of the ring, associating with PIH1D1, which mediates client protein recruitment. Thus, RPAP3 provides a flexible platform for bringing HSP90 into proximity with diverse client proteins.

[1] Centro de Investigaciones Biológicas (CIB), Spanish National Research Council (CSIC), Ramiro de Maeztu 9, 28040 Madrid, Spain. [2] Genome Damage and Stability Centre, School of Life Sciences, University of Sussex, Falmer, Brighton BN1 9RQ, UK. [3] Spanish National Cancer Research Centre (CNIO), Melchor Fernández Almagro 3, 28029 Madrid, Spain. [4] Structural Biology Unit, CIC bioGUNE, Bizkaia Technology Park, 48160 Derio, Spain. [5] MRC Laboratory of Molecular Biology, Cambridge Biomedical Campus, Francis Crick Avenue, Cambridge CB2 0QH, UK. These authors contributed equally: Fabrizio Martino, Mohinder Pal, Hugo Muñoz-Hernández, Carlos F. Rodríguez. Correspondence and requests for materials should be addressed to L.H.P. (email: Laurence.Pearl@sussex.ac.uk) or to O.L. (email: ollorca@cnio.es)

The R2TP/Prefoldin-like (R2TP/PFDL) complex collaborates with the HSP90 molecular chaperone to facilitate assembly, activation, and cellular stability of a range of multiprotein complexes, including RNA polymerase II (Pol II), complexes of PI3 kinase-like kinases (PIKKs) such as TOR and SMG1, and small nuclear ribonuclear protein (snRNPs) complexes, amongst others[1–7]. Yeast R2TP complexes comprise four subunits, RuvB-like AAA+ ATPases Rvb1p and Rvb2p, a TPR domain-containing protein Tah1p, and a PIH domain protein Pih1p. Metazoan R2TP complexes contain the orthologous proteins RUVBL1, RUVBL2, RPAP3, and PIH1D1, respectively. However, whereas the TPR domain-containing component of the yeast R2TP complex is a small (12 kDa) protein, Tah1p, in human R2TP this is a large (75 kDa) multi-domain protein, RPAP3 (RNA polymerase II associated protein 3) (or hSPAGH), containing two TPR domains. The C-terminal region in RPAP3 has been annotated as a protein domain (pfam13877), which is also present in other proteins, such as CCDC103[8], a dynein arm assembly factor that interacts with RUVBL2[9].

In mammals, the R2TP core components associate with additional subunits of the prefoldin (PFDL) module, forming the R2TP/PFDL complex. This PFDL module includes prefoldin and prefoldin-like proteins PFDN2, PFDN6, URI1, UXT, PDRG1, and it associates with two additional components, the RNA polymerase subunit POLR2E/RPB5 and WDR92/Monad[5,10]. In addition, R2TP/PFDL interacts with additional proteins that serve as adaptors between R2TP/PFDL and the clients (see later)[5,11,12].

RPAP3 was first identified and named after a systematic analysis of complexes containing components of the transcription and RNA processing machineries using protein affinity purification coupled to mass spectrometry[13]. RPAP3 was then found to be a component of the multi-subunit R2TP/PFDL complex[14]. Subsequently it was found to associate with Pol II subunits and HSP90 when Pol II assembly is blocked by α-amanitin, implicating both RPAP3 and HSP90 in Pol II assembly in the cytoplasm[10]. Pol II subunits RPB1, RPB2, and RPB5 all co-precipitate with RPAP3, but RPAP3 seems to associate independently with RPB1 and RPB5-containing complexes, suggesting the existence of different RPAP3 complexes as intermediates in Pol II assembly. RPAP3 also binds some subunits of RNA Pol I and it may therefore play a more general role in the assembly of all RNA polymerases[10]. The mechanistic details of how RNA Pol II subunits are recruited to R2TP and how R2TP and HSP90 contribute to Pol II assembly are poorly understood. Unconventional prefoldin RPB5 Interactor 1 (URI1) interacts with the RPB5/POLR2E subunit of Pol II, and this suggests that the PFDL module contributes to recruit Pol II assembly intermediates to the R2TP/PFLD complex[10,15].

Recruitment of PIKK proteins to R2TP is mediated by the phosphopeptide-binding PIH domain at the N-terminus of Pih1p/PIH1D1, which recognizes a specific phosphorylated acidic motif, generated by casein kinase 2 (CK2)[2–4]. This motif is conserved in Tel2p/TELO2, a component of the TTT (Tel2p/TELO2–Tti1p/TTI1–Tti2p/TTI2) complex that also interacts directly with PIKKs, thereby bridging their interaction to R2TP. A similar PIH-binding motif is also found in Mre11p/MRE11A suggesting that R2TP may also play a role in the assembly of MRN complexes involved in DNA double-strand break repair[2]. Neither Pol II nor snRNPs subunits contain this motif, and must therefore be recruited to R2TP through alternative mechanisms. Biogenesis of box C/D snoRNP requires R2TP and additional factors such as NUFIP1 and the Zinc-finger HIT domain proteins ZNHIT3 and ZNHIT6, which have been proposed to function as adaptors between R2TP and C/D core proteins[12]. Interestingly, ZNHIT2, another protein of the same family, was recently shown to bind RUVBL2 and regulate assembly of U5 small

ribonucleoprotein[5]. ZNHIT2 may function as a bridging factor between the U5 snRNP and the R2TP/PFDL, a function where the Ecdysoneless (ECD) protein could also contribute[5,16]. Human ECD homolog interacts with the pre-mRNA-processing-splicing factor 8 (PRPF8)[17], and the R2TP[18]. Phosphorylated ECD interacts with the PIH1D1 subunit, as well as with RUVBL1 in a phosphorylation-independent manner[18]. Therefore, it seems that sets of different adaptors collaborate to bring specific clients to R2TP/PFLD.

Previous structural and biochemical studies have defined most of the pairwise interactions of R2TP core components. The TPR domain of yeast Tah1p mediates interaction with the conserved MEEVD C-terminal tail peptide of HSP90[2,19–22], while its C-terminal extension couples Tah1p to the CS-domain of Pih1p[2,21]. The central region of Pih1p mediates recruitment of Pih1p–Tah1p to the Rvb1p–Rvb2p heterohexameric ring[23,24]. The N-terminal PIH domain of Pih1p/PIH1D1 binds a CK2-phosphorylation motif on Tel2p/TELO2, mediating recruitment of the TTT complex to R2TP[2,3]. Most recently, we have determined the cryo-EM structure of the intact yeast R2TP complex, in which a single Tah1p–Pih1p sub-complex binds a heterohexameric Rvb1–Rvb2 ring[24], a finding subsequently confirmed by others[25]. In metazoan R2TP, the small (12 kDa) single-TPR domain protein Tah1p is replaced by the much larger (75 kDa) RPAP3/hSpagh whose N-terminal half contains a tandem pair of TPR domains that bind in concert to a single HSP90 dimer[2]. However, the function of the rest of RPAP3 is unknown. To our knowledge, the subunit stoichiometry and the structural organization of a metazoan R2TP complex have not been determined.

To gain further insight into how R2TP/PFDL functions in the assembly, activation and stabilization of its 'client' systems, we have determined the cryo-EM structure of human R2TP core complex. Our data reveal a substantially elaborated architecture compared with the yeast system, in which RPAP3 rather than PIH1D1 plays the central organizational role, incorporating additional domains and functions to address the assembly of a variety of large complexes. We identify the C-terminal domain in RPAP3 as a helical bundle that binds selectively to the ATPase domain of RUVBL2. As well as scaffolding the interaction of PIH1D1 with the RUVBL1–RUVBL2 ring, RPAP3 provides a flexible tether for HSP90, allowing it to interact with a highly diverse set of client proteins and complexes.

## Results

**Recruitment of R2TP components by RPAP3.** The human TPR domain protein RPAP3 is roughly six times larger than its yeast equivalent Tah1p, and we sought to determine whether it may provide docking sites for other components of the human R2TP complex (Fig. 1a). Yeast Pih1p constructs containing the C-terminal CS domain, and the isolated CS domain itself, are unstable in isolation, but are stabilized by interaction with the C-terminal tail of Tah1[26]. We found that human PIH1D1 protein was also unstable when expressed in isolation, and we used this property to identify a minimal PIH1D1 binding motif in RPAP3 by co-expressing PIH1D1 with GST-tagged RPAP3 constructs and looking for co-purification of PIH1D1 in GST pull-downs from cell lysates. As well as in the full-length GST-RPAP3, we found that constructs that contained residues 400–420 of RPAP3, immediately downstream of the second RPAP3 TPR domain, were able to form a stable and soluble complex with full-length PIH1D1 or its isolated CS domain, when co-expressed (Fig. 1b). PIH1D1-CS was not co-purified when co-expressed with a GST-RPAP3 construct lacking residues 400–420 (Fig. 1c). We conclude that residues 400–420 of RPAP3 and the CS domain of PIH1D1 are together both necessary and sufficient to mediate the interaction of the two proteins.

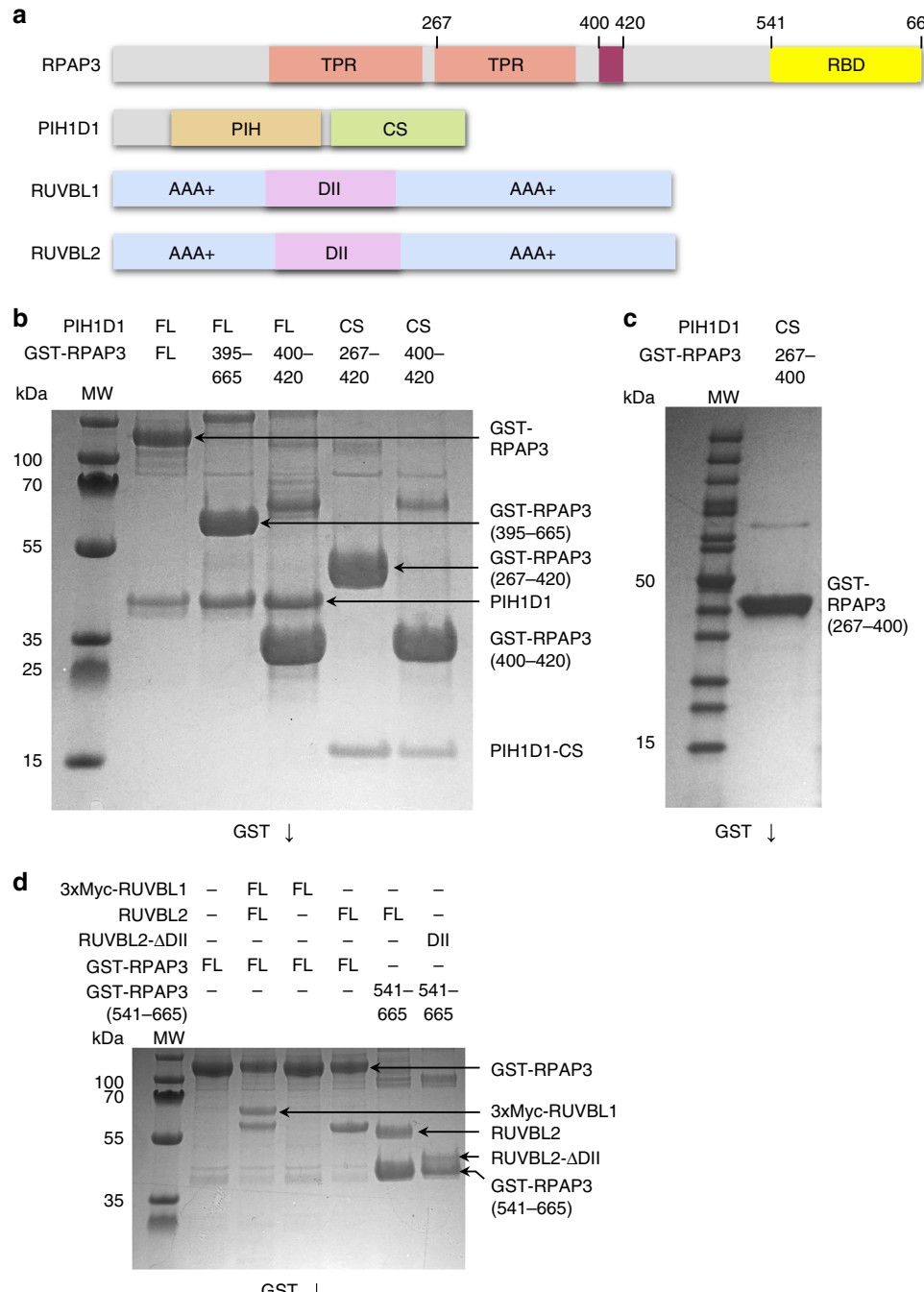

**Fig. 1** Mapping the interactions in human R2TP core components. **a** A cartoon for sequence and domains of the components of the human R2TP complex. **b** GST pull-down experiments depicting the interactions between the several regions in RPAP3 and PIH1D1. FL stands for full length, CS for the CS domain in PIH1D1, and MW for molecular weight markers. Be aware that for simplification, several PIH1D1 and RPAP3 constructs are indicated within the same lines on top of the gel. Some minor contaminants are present in some of the samples. **c** Pull-down experiments showing that removal of residues 401–420 from an RPAP3 construct eliminates the interaction with the CS domain in PIH1D1. **d** Pull-down experiments demonstrating the interaction of RPAP3–RBD with RUVBL2. This interaction is not affected when the DII domains in RUVBL2 are removed

We found that full-length RPAP3 protein in the absence of PIH1D1 was fully competent to bind to the assembled RUVBL1–RUVBL2 heterohexamer but it binds RUVBL2 and not RUVBL1 when they are not forming a complex, suggesting that it is RUVBL2 that mediates most of the interactions to recruit RPAP3 (Fig. 1d). Dissection analysis of RPAP3 identified a segment of the polypeptide between Valine 541 and Glycine 665 as necessary and sufficient to bind RUVBL2 (hereinafter referred to as RBD, RUVBL2-Binding Domain). The RPAP3–RBD was also able to pull-down an RUVBL2 construct lacking the DII 'insertion' domain (RUVBL2-ΔDII), suggesting the RPAP3–RBD domain interacts with the ATPase domain face of RUVBL2 rather than the DII domain face implicated in dodecamer formation (Fig. 1d). The RBD is the only domain essential to maintain the RPAP3–RUVBL2 interaction, since an RPAP3–PIH1D1 complex where the RBD is truncated did not bind RUVBL2 (Supplementary Fig. 1). An N-terminal 3xMyc tag in RUVBL1 was used to allow RUVBL1 and RUVBL2 to be discriminated in

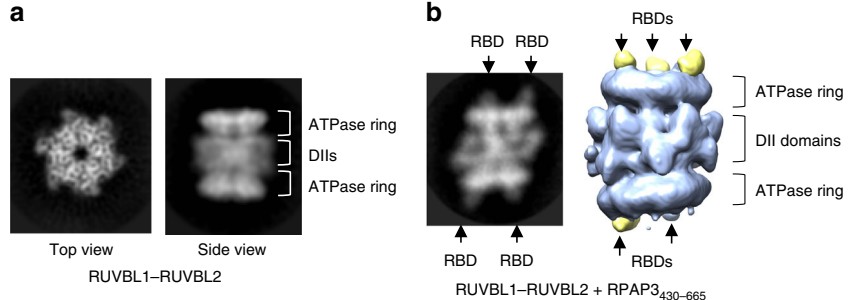

**Fig. 2** Cryo-EM imaging of RUVBL1–RUVBL2 and the RBD domain. **a** 2D averages corresponding to top and side views obtained from cryo-EM images of the RUVBL1–RUVBL2 preparation in an ADP-containing buffer. **b** After incubation with RPAP3$_{430-665}$, RBDs decorate the ATPase side of both RUVBL rings without disrupting the dodecamer, and a representative 2D average of the complex between RUVBL1–RUVBL2 and RBD is shown. At the right end of the panels, one view of the 3D structure of RUVBL1–RUVBL2–RBD complex with RBD domains in yellow. Note that one of the RBD domains in the bottom ring is less visible at the threshold used for rendering, probably reflecting variable occupancy. Also, the scale of the 3D structure has been enlarged with respect to the 2D average, for clarity

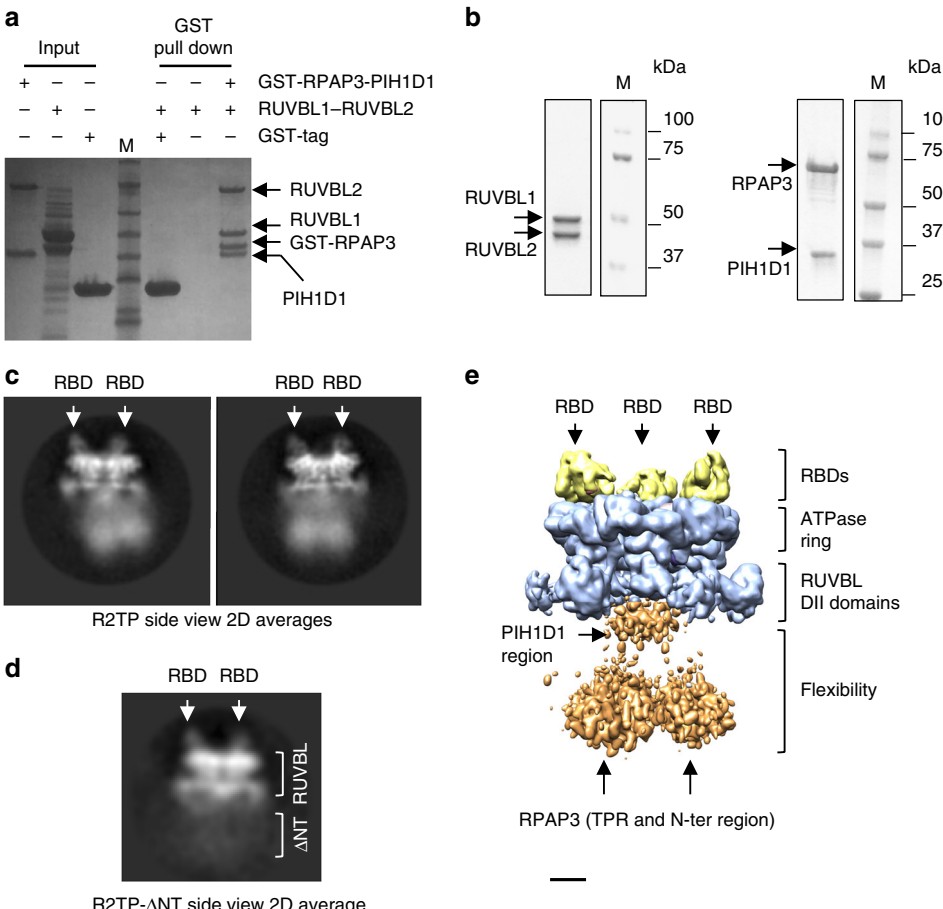

**Fig. 3** Cryo-EM imaging of the R2TP complex. **a** Pull-down experiments showing the in vitro reconstitution of R2TP. M indicates molecular weight markers. **b** Purification of RUVBL1–RUVBL2 and PIH1D1–RPAP3 sub-complexes, used for the reconstitution of R2TP for cryo-EM. M indicates molecular weight markers. **c** Two representative side view averages of R2TP. RUVBL1–RUVBL2 rings are decorated by the RBD at the top (labeled with white arrows). A blurred and very flexible region locates at the bottom of the ring. **d** A representative side view average of R2TP reconstructed using the RPAP3–ΔNT–PIH1D1 sub-complex and RUVBL1–RUVBL2. Flexible regions at the bottom end of R2TP disappear when the N-terminal half of RPAP3 is removed, but dodecameric RUVBL1–RUVBL2 is disrupted. **e** 3D structure of R2TP obtained applying 3-fold symmetry. RBDs are bound to RUVBL1–RUVBL2 but the flexible regions in the complex are not resolved. Scale bar, 2.5 nm

sodium dodecyl sulfate-polyacrylamide gel electrophoresis (SDS-PAGE). The N-terminal end of RUVBLs locates at the DII domain face, and thus the tag is unlikely to affect binding of the RBD to the ATPase domain face. Control experiments rule out

any deleterious effects of the GST tag and the tag in RUVBL1 (Supplementary Fig. 1).

Together, our interaction mapping reveals a very different assembly of the PIH domain and TPR domain-containing

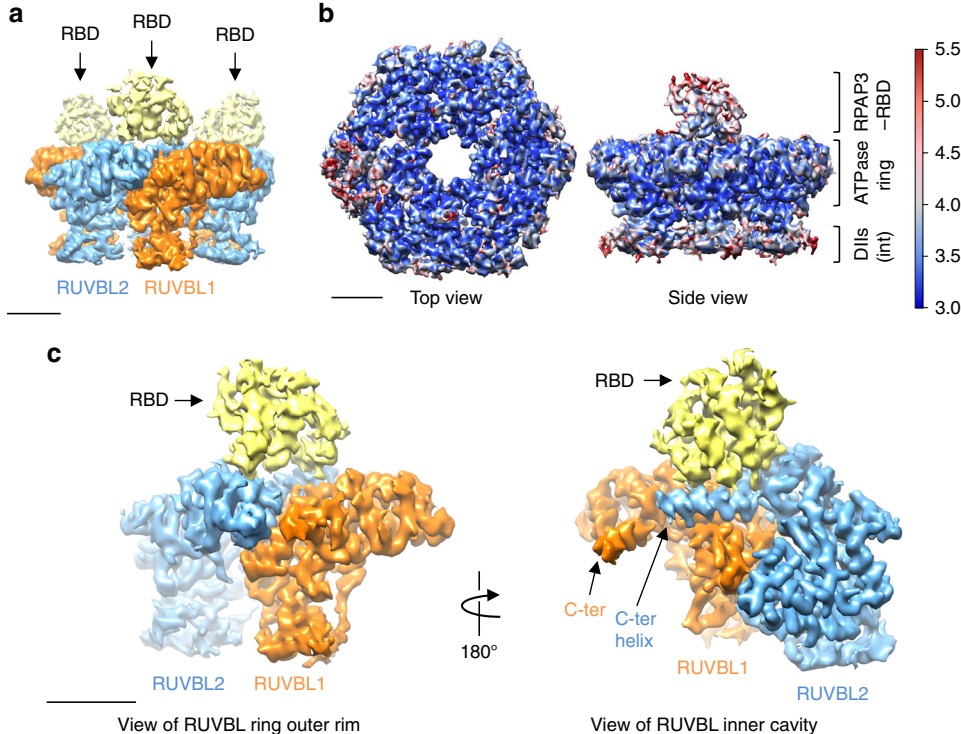

**Fig. 4** Cryo-EM structure of RUVBL1–RUVBL2–RBD. **a** Top and side view of the cryo-EM density for the RUVBL1–RUVBL2–RBD complex containing 3 RBDs. RUVBL1 is colored in orange, RUVBL2 in blue, and RBD in yellow. Scale bar, 2.5 nm. **b** Top and side view of the 3.6 Å resolution structure of the RUVBL1–RUVBL2–RBD complex, processed as indicated in Methods, and displaying 1 RBD per RUVBL ring, and colored according to resolution difference from 3.0 to 5.5 Å[46]. Scale bar, 2.5 nm. **c** Two views of a 1:1:1 RUVBL1–RUVBL2–RBD complex, as seen from the inside or the outside of the RUVBL1–RUVBL2 ring. The position of the C-terminal helices of RUVBL1 and RUVBL2 are indicated. Scale bar, 2.5 nm

components of human R2TP than in the yeast system[24]. Instead of Pih1p acting as the central scaffold that connects the HSP90-recruitment factor Tah1p to the AAA+ ring, RPAP3 takes at least part of this role, interacting with HSP90, RUVBL1–RUVBL2 and PIH1D1, and providing additional domains, which may mediate recruitment of other factors to the R2TP core[27]. Most significantly, the primary interaction between the TP component and the RUVBL1–RUVBL2 heterohexamer in human R2TP, is mediated by a RUVBL2-binding domain located at the C-terminus of RPAP3, corresponding to the previously annotated protein domain (pfam13877).

**RPAP3–PIH1D1 but not RBD disrupts dodecameric RUVBL1–RUVBL2.** The cryo-EM images of RUVBL1–RUVBL2 alone, obtained using the same experimental conditions as those used later for the full R2TP, showed that they exist as a back-to-back dodecameric complex with the DII domains mediating the interaction between two hexamers, and the ATPase rings facing outward (Fig. 2a), as previously described[28]. In all cases, we used Adenosine 5′-diphosphate (ADP) in the buffer, which stabilizes RUVBL1–RUVBL2-containing complexes[29,30]. When RUVBL1–RUVBL2 was incubated with a fragment of RPAP3 comprising residues 430–665, up to 3 RBDs decorated the ATPase face of the RUVBL ring, one per each RUVBL2 molecule in the complex (Fig. 2b). In our conditions, one ring in the dodecamer was saturated whereas the other contained a variable number of RBDs, indicating we had not reached saturation. The location of the RBD at the ATPase face was consistent with its binding to RUVBL2-ΔDII in pull-down experiments (Fig. 1d) and with cross-links detected between the RBD and residues K453 of RUVBL1 and K417 in RUVBL2, both exposed at the ATPase

face of the RUVBL ring (Supplementary Fig. 2). These experiments show that the interaction of RPAP3 residues 430–665, which include the RBD, with RUVBL1–RUVBL2 was insufficient to disrupt the dodecamer.

R2TP could be reconstituted by mixing RUVBL1–RUVBL2 and RPAP3–PIH1D1 (Fig. 3a). For cryo-EM studies, R2TP was assembled from RUVBL1–RUVBL2 and RPAP3–PIH1D1 sub-complexes, each co-expressed and purified by affinity and gel-filtration chromatography (Fig. 3b). Images of the fully assembled human R2TP complex revealed a single hexameric ring of RUVBL1–RUVBL2 in which its ATPase face was decorated with the RBD domain of RPAP3 (Fig. 3c, Supplemental Fig. 3). The remaining regions of RPAP3 appeared as blurred density at the opposite side of the RUVBL1–RUVBL2 ring, tilted with respect to the ring, indicative of substantial structural flexibility in their connection to the core of the complex.

Flexible regions at the opposite end of the RUVBL ring are attributed to the N-terminal TPR-containing region of RPAP3. In support of this, this density is absent from the cryo-EM images when R2TP was reconstituted with a truncated version of RPAP3 (residues 395–665) bound to PIH1D1, but from which the TPR domains and N-terminus of RPAP3 have been removed (RPAP3-ΔNT) (Fig. 3d). These images suggested that PIH1D1 located close to the RUVBL ring, as in yeast[24,25]. Therefore, RPAP3–PIH1D1 and RPAP3–ΔNT–PIH1D1 are sufficient to disrupt the RUVBL1–RUVBL2 dodecamers. Since RPAP3 residues 430–665 bind the RUVBL ring without affecting the dodecameric assembly of RUVBL1–RUVBL2, our data suggest that the RPAP3 region that binds PIH1D1 together with PIH1D1 itself are responsible for these effects, likely because it locates within the DII-domain face of the RUVBL ring, as in yeast[24]. This is the side of the RUVBL ring where most protein interactions have been described in other complexes that contain a

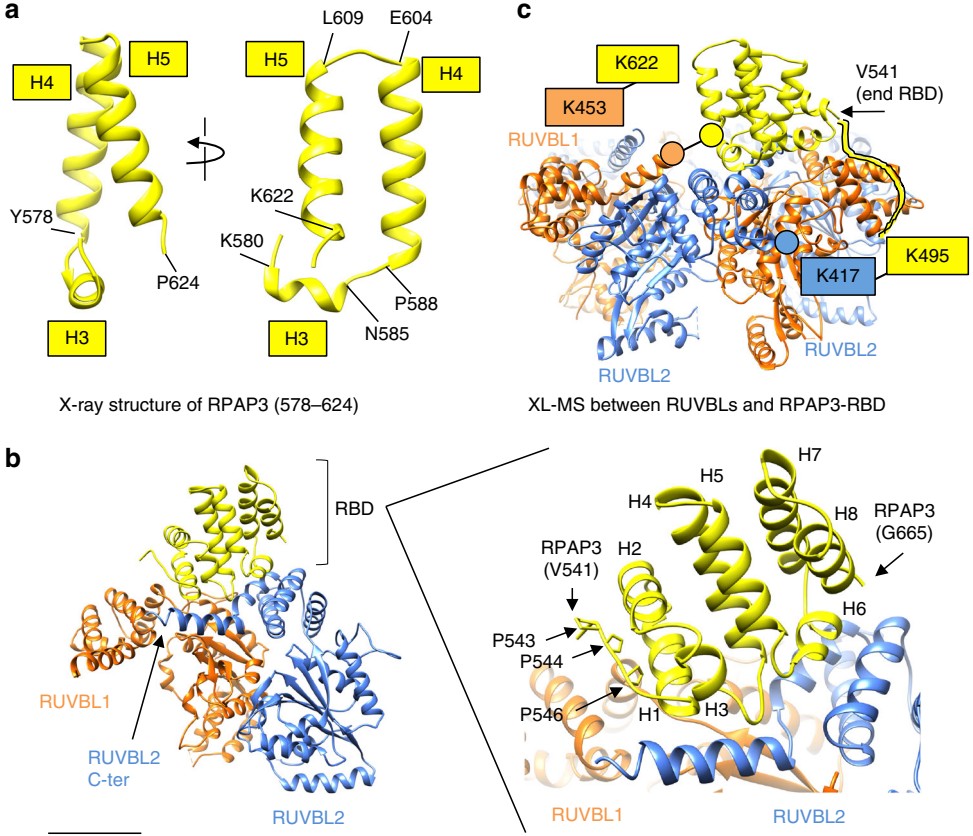

**Fig. 5** Structural model of RUVBL1–RUVBL2–RBD. **a** Crystal structure of a fragment of the RBD domain. H3, H4, and H5 stand for helices 3, 4, and 5, respectively. **b** Structural model of RUVBL1-RUVBL2-RBD. Color codes: RUVBL1 (orange color), RUVBL2 (blue color), and the RBD (yellow color). The panel on the left shows only two subunits from RUVBL1-RUVBL2 as seen from the interior of the ring. The panel on the right is a close-up view of the RBD to highlight the N-terminal region and the positioning of the 8 helices in the structure (H1 to H8). Scale bar, 2.5 nm. **c** Two cross-links identified by XL-MS between RUVBL1-RUVBL2 and the RBD are indicated on the structural model

RUVBL1–RUVBL2 hexamer, such as INO80[31]. These experiments do not discard that RPAP3 alone could be sufficient to disrupt the RUVBL1–RUVBL2 dodecamer.

**R2TP can engage up to 3 RPAP3 molecules**. R2TP images were classified according to the number of RBDs per ring, by masking out the rest of the molecule and without assuming symmetry (Supplementary Fig. 3). This classification revealed that most of the RUVBL rings contained 3 RBDs (>67%), suggesting that R2TP can incorporate up to 3 RPAP3 molecules, one for each RUVBL2 in the complex. This was also the subgroup where the RBDs displayed better quality in the cryo-EM images. It is noteworthy that several orientations of R2TP complexes containing 3 RBDs may seem to contain only 2 apparent RBDs in the two-dimensional (2D) averages (Fig. 3c), but this is because 2 RBDs coincide in the same direction of the projection for many of the most abundant rotations of R2TP along its longitudinal axis in our data set, thus masking each other (Supplementary Fig. 3).

Images of the most abundant, RBD-saturated complex were then processed applying 3-fold symmetry (Fig. 3e). The structure revealed a RUVBL1–RUVBL2 hexamer decorated by 3 RBDs. However flexible regions located at the opposite end of the RUVBL ring did not follow the 3-fold rotational symmetry. These regions included the TPR end of RPAP3, which was mapped using RPAP3-ΔNT (Fig. 3d), and also a density in the DII-face of the RUVBL ring that should contain PIH1D1. The rigid and flexible segments of R2TP were processed separately with dedicated image processing strategies (see later).

**Cryo-EM structure of the RUVBL1–RUVBL2–RBD complex**. The structure of RUVBL1–RUVBL2–RBD was first resolved using 3-fold symmetry, revealing a helical domain bound to each RUVBL2 in the hexameric ring (deposited as R2TP-C3 symmetry in EMDB) (Fig. 4a). When the processing was performed without symmetry, small differences in the quality of each RBD in the ring were observed. Thus, to attain maximum resolution of the RUVBL1–RUVBL2–RBD interaction, particles were rotated by 120 and 240° so that each of the 3 RBDs in every particle was placed in the same position, allowing the classification of all the available RBD data (Fig. 4b, c). Similar methodology was applied previously by others to resolve the cryo-EM structure of the apoptosome[32] (see details in Methods). Three-dimensional (3D) classification generated subgroups corresponding to small differences in some of the α-helices of the RBD, and the one that reached the best resolution was refined locally to an estimated average resolution of 3.6 Å (Supplementary Fig. 3). Analysis of the local resolution revealed that most of the map showed resolutions between 3.0 and 3.5 Å, with some external parts ranging between 3.5 and 5 Å (deposited as R2TP–1RBD in EMDB) (Fig. 4b). The RBD structure revealed a α-helical domain sitting on top of each RUVBL2 subunit, projecting out from the ATPase domain face of the RUVBL1–RUVBL2 ring (Fig. 4c).

Previously described interactions with RUVBL1–RUVBL2 such as in SWR1[33], INO80[11], or yeast R2TP[24,25] are mediated by the face of the ring presenting the DII 'insertion' domain. Interaction with the opposite ATPase domain face of the RUVBL1–RUVBL2, as seen here with RPAP3-RBD, has not previously been found,

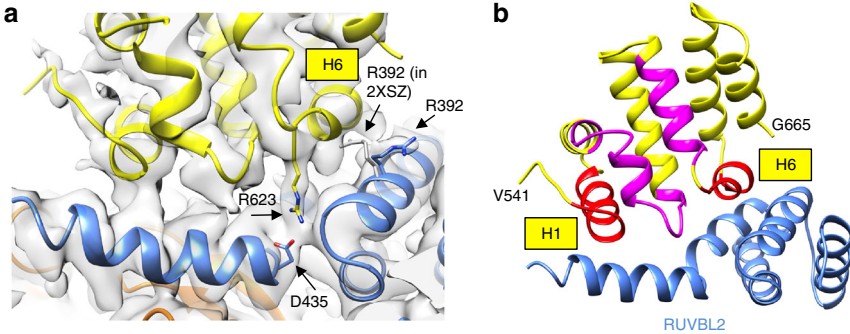

**Fig. 6** Structural details of RUVBL1–RUVBL2–RBD. **a** R392 residue in RUVBL2 of the RUVBL1–RUVBL2–RBD complex (model in blue color within the density) shows a different conformation to its position in the crystal structure of RUVBL1–RUVBL2 (structure in gray color) (PDB 2XSZ)[29]. R623 in RBD is also visible and pointing toward RUVBL2. **b** Helix 1 and 6 are 100% identical in those species analyzed (red color). Other regions shown in magenta color also revealed high conservation

although contacts between this face of the RUVBL ring have been observed in crystals of Rvb1/Rvb2 dodecameric complexes[34].

**RPAP3–RBD recognizes specific features in RUVBL2.** Side chains and other high-resolution details are clearly visible in the RUVBL1–RUVBL2 component of the map (Supplementary Fig. 4), such as the ADP density and the surrounding side chains, allowing an accurate atomic modeling of these two proteins in the complex. As no experimental crystallographic or Nuclear magnetic resonance (NMR) structure has been reported for the RPAP3-RBD, a structural model was generated using the I-TASSER server[35], which predicted that residues 541–665 comprise 8 highly conserved α-helices preceded by a long and poorly structured region (Supplementary Fig. 4). Attempts to crystallize the RBD resulted in a crystal structure at 1.8 Å resolution of a proteolytic fragment comprising residues 578–624 (deposited as 6FM8 in PDB). This fragment accounted for a third of the RBD domain approximately, including most of helix 3, and the complete helices 4 and 5 (Fig. 5a). The crystal structure supported assignment of the correct register of amino acids, and it helped to model the connection between helices 5 and 6. The conformation of such a short segment will be strongly influenced by crystal packing, and thus the overall conformation of the RBD observed in cryo-EM was considered for the model. The disposition of secondary structure elements in the prediction clearly matched the density for the RBD we observed in the cryo-EM, and the model was then flexibly fitted within the map (see Methods for details) (Supplementary Movie 1). The fitted structure of the RBD and the crystal structure of the RUVBL1–RUVBL2 hetero-hexamer (PDB 2XSZ)[29] were refined in Phenix.refine and adjustments made in COOT[36]. The atomic model obtained (Fig. 5b) showed good cross-correlation with the cryo-EM map (Supplementary Fig. 4). Density for side chains were also visible in most of the helices of the RBD (Supplementary Fig. 5), and this, together with the crystal structure of the RBD fragment, were used to tune and validate the fitting of the atomic model into the cryo-EM map. Nonetheless, some side chains in the interaction surface between RUVBL1–RUVBL2 and RBD are not well defined, and thus detailed protein–protein interactions are mostly discussed at the level of secondary structural elements, except where side chain density was clear.

We subjected the assembled complex to analysis by cross-linking mass spectrometry (XL-MS) (Supplementary Fig. 2). XL-MS identified cross-links between residue K453, a lysine in RUVBL1 C-terminal helix, and K622 in RPAP3, and between K417 in RUVBL2 and K495 in RPAP3—both cross-links are compatible with the structure (Fig. 5c). Although the bound RPAP3-RBD is proximal to some regions in RUVBL1 and an RPAP3–RUVBL1 cross-link was observed, as expected from the pull-down data, the bulk of the RPAP3 interaction with the AAA+ ring occurs with RUVBL2, and is mediated by selective interaction with conserved features that are absent in RUVBL1 (Fig. 5b). As a control during the adjustment of the RBD model into the cryo-EM map, we found that any fitting in the reverse N- and C-terminal orientation did not agree with the cryo-EM density but, in addition, was incompatible with the XL-MS data.

RPAP3–RBD is comprised of 8 helical segments (H1 to H8), and they make contacts with RUVBL2 through H1, H6, and the loop connecting H5 and H6 (Fig. 5b). Although no secondary structure is predicted for RPAP3 immediately N-terminal of H1, the electron density for this segment was good enough to model a loop between Leu541 and Asn548 based on the presence of three prolines (Pro543, Pro544, Pro546) (Fig. 5b). This loop leads toward the outer edge of the RUVBL1–RUVBL2 ring (see later). H1 contacts the C-terminal region of RUVBL1 (Fig. 5b), whereas H6 sits between 2 helices of RUVBL2 that are positioned in a V-shape fashion (Figs. 5b, 6a). The V-shaped fold of RUVBL2 that accommodates H6 of the RPAP3–RBD is rich in positively charged amino acids such as Arg392, which is moved away from its location in the crystal structure of RUVBL1–RUVBL2 alone (PDB 2XSZ)[29] (Fig. 6a), suggesting that it may be repositioned to avoid clashing and/or to make contacts with the RBD. Density for the side chain of Arg623 of the RBD is also well defined, pointing toward residues in RUVBL2, with which it may establish contacts (Fig. 6a).

The RPAP3–RBD is strongly conserved in chordates, and the amino acid sequences of H1 and H6 are identical in humans, rats, mice, chicken, and *Xenopus* (Fig. 6b, Supplementary Fig. 4, 100% identical regions labeled in red color), highlighting its functional importance, most likely for the interaction with RUVBL2. The regions of RUVBL2 that interact with RPAP3-RBD are also very well conserved, but these are also conserved in yeast suggesting that RPAP3, and not RUVBL2, evolved to exploit the characteristics of this region in RUVBL2.

**RPAP3 loops around the RUVBL1–RUVBL2 hexamer.** Cryo-EM images of R2TP suggested that the RPAP3 molecules span both faces of the RUVBL1–RUVBL2 ring (Fig. 3c). This would be facilitated by the long unstructured region (residues 420–540) that connects the tandem TPR domains and PIHID1-binding region of RPAP3 to the C-terminal RBD (Fig. 1a). In an extended conformation, this segment could comfortably stretch over several nanometers, allowing the RBD to bind on one face of RUVBL1–RUVBL2, while the rest of RPAP3 is located on the

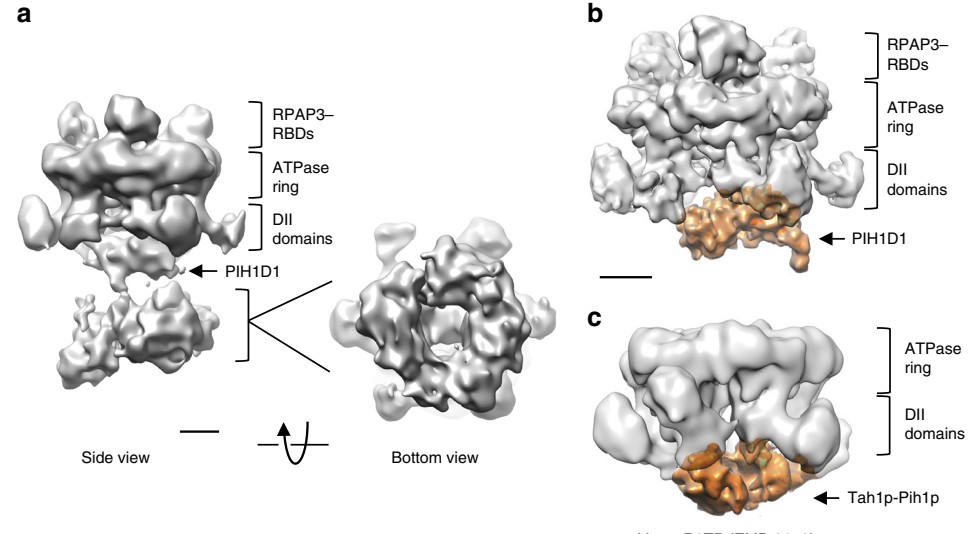

**Fig. 7** Structural analysis of the flexible region in R2TP. **a** Side and bottom view of the output 3D classification for a subset of particles with a more homogenous conformation for the flexible regions. Scale bar, 2.5 nm. **b** Low-resolution structure of PIH1D1 region in R2TP complexes. Density for PIH1D1, highlighted in orange color. Scale bar, 2.5 nm. **c** One view of the structure of yeast R2TP (EMD 3678)[24]. Rvb1–Rvb2 is shown in white transparency whilst the density for the yeast Tah1p–Pih1p complex is shown in orange color

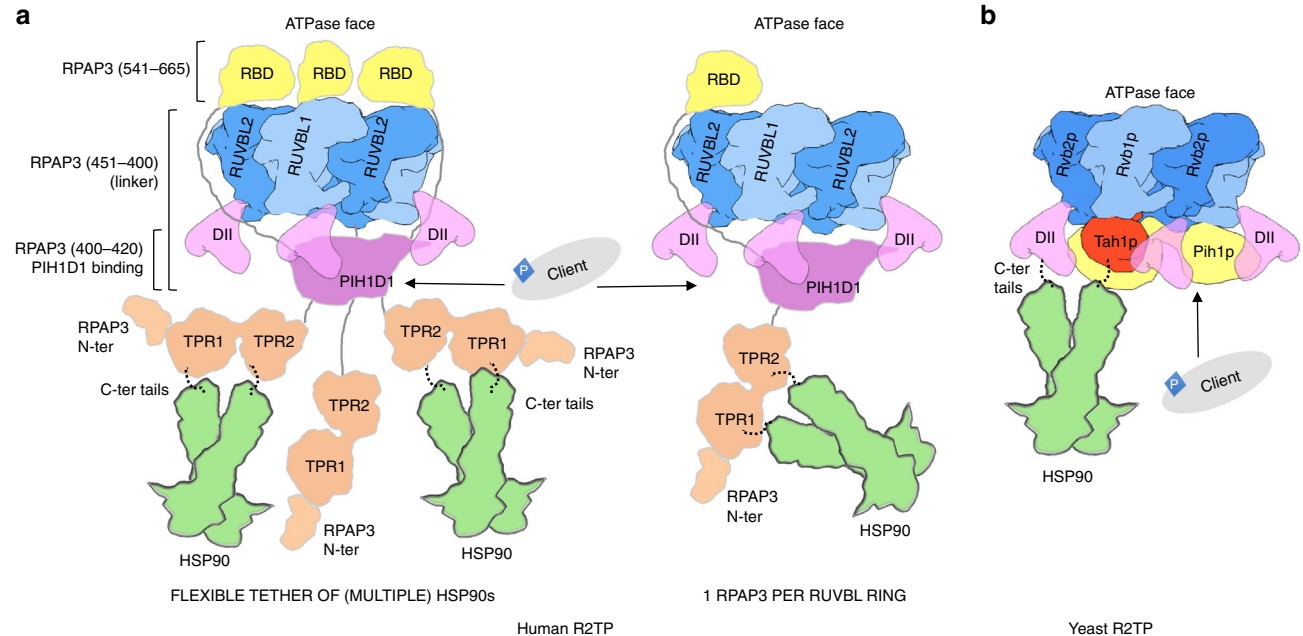

**Fig. 8** A cartoon for the structural and functional model for R2TP. **a** Human R2TP. HSP90 dimers can engage with each R2TP complex with sufficient conformational flexibility to reach and act in a diversity of client proteins. Up to 3 RBDs serve to anchor 3 RPAP3 to the RUVBL1–RUVBL2 scaffold, whereas a central segment of RPAP3 helps to recruit PIH1D1. The number of RPAP3 molecules per RUVBL ring in vivo is not known, and two options are shown in the figure. A long and poorly structured link between the RBD and TPR domains in RPAP3 results in substantially conformational flexibility of the TPR regions. For simplicity, although 3 RBDs are bound to the RUVBL ring, only 2 RPAP3s are shown bound to HSP90 in the cartoon. **b** Yeast R2TP. Conformational adaptability of yeast R2TP is limited to the flexibility of the C-terminal tails in Hsp90. Only one Hsp90 binds each R2TP.

opposite face of the ring. The visible N-terminal loop on the RBD (Leu541–Asn548) comes in from the edge of the ATPase domain face (Fig. 5b), suggesting that the preceding polypeptide chain runs across the rim of the RUVBL1–RUVBL2 ring. Consistent with such a trajectory, we identified a cross-link between RPAP3-Lys495, which lies within the unstructured region, and RUVBL2-Lys417, which projects from the rim of the RUVBL1–RUVBL2 ring (Fig. 5c).

**RPAP3 provides a flexible tether for HSP90.** Human R2TP was fully competent to bind HSP90 and this interaction is mediated by the TPR domains in the N-terminal half of RPAP3 (Supplementary Fig. 6). Unlike the RBD, which is rigidly bound, this region is disordered relative to the RUVBL1–RUVBL2 (Fig. 3). Two image-processing strategies were used to define the structure of the flexible regions in R2TP comprising the TPR domains of RPAP3 and PIH1D1.

Extensive classification was used to select a subset of particles displaying a more homogenous conformation in the flexible regions (deposited as R2TP-subgroup1 in EMDB) (Fig. 7a). In this structure, several compact densities were resolved on the DII domain face of the RUVBL ring. Densities at the further end of the complex, opposite to the RUVBL ring, had been assigned as comprising the N-terminal region of RPAP3 and the TPR domains (Fig. 3d), and dimensions were sufficient to accommodate the TPR domains of 3 RPAP3 subunits. This interpretation was further supported by images of R2TP reconstituted using N-terminally GST-labeled RPAP3, where the additional density for the GST fusion mapped to the face of the RUVBL ring opposite to that bound by the RBD (Supplementary Fig. 6). On the other hand, density for PIH1D1 located within the DII region beneath the RUVBL ring.

To analyze the region around PIH1D1, images corresponding to R2TP complexes containing 3 RBDs were classified and aligned masking out flexible regions except for the vicinities of the RUVBL1–RUVBL2 ring, and image processing was performed without applying any symmetry. This approach resolved a clear region of density within the cage formed by the DII domains in the RUVBL ring (deposited as R2TP-subgroup2 in EMDB) (Fig. 7b), in a similar position to that occupied by Pih1p and Tah1p in yeast R2TP[24] (Fig. 7c). As in the yeast system, the inherent flexibility of this region did not permit a sufficiently high resolution to clearly define secondary structure elements.

## Discussion

HSP90 is required for the stabilization, activation, and assembly of a diverse range of proteins and complexes involved in cellular processes as fundamental and as varied as transcription, cell cycle progression, centrosome duplication, telomere maintenance, siRNA-mediated gene silencing, apoptosis, mitotic signal transduction, innate immunity, and targeted protein degradation[37]. HSP90s ability to chaperone this very broad protein clientele is provided by co-chaperones[38], which act as adaptors, facilitating recruitment of client proteins to HSP90.

R2TP/PFDL is the most complex HSP90 co-chaperone yet described, and is known to facilitate HSP90s participation in the assembly of RNA polymerases, PIKK complexes, and snoRNPs[39,40], although this list is likely to be far from complete. It recruits HSP90 via its TPR domain component Tah1p or RPAP3, and at least some of its clientele through a CK2 phosphorylation-dependent interaction with the PIH domain of Pih1p/PIH1D1[2,3]. The best characterized of these interactions involves Tel2p/TELO2, itself a component of an additional 'adaptor' layer, the TTT complex (Tel2p/TEL2, Tti1p/TTI1, and Tti2p/TTI2), which ultimately bridges R2TP, and thereby HSP90, to PIKK proteins[7,41,42]. Comparable PIH-binding motifs can be identified in a range of other proteins in yeast and metazoa, suggesting that there are many R2TP-mediated HSP90 dependencies yet to be described, some of which will likely also involve multiple adaptor systems.

The structure of the human R2TP core components revealed here is well suited to deal with a high degree of diversity in the client proteins it brings to HSP90 (Fig. 8). In both yeast and human R2TP, a PIH domain client-recruitment component maps to the same face of the RUVBL ring as the TPR domain(s) necessary for HSP90 recruitment, facilitating direct interaction of the chaperone and the client (or at least client adaptor).

In the yeast system, the single TPR domain of Tah1p is part of a well-ordered cluster of domains along with the CS and PIH domains of Pih1p, that lies in the cup formed by the DII domains of the RUVBL ring[24,25]. Conformational adaptability in the yeast system is thus limited to the inherent flexibility of the ~30

residues linking the last globular domain of the bound HSP90 to the TPR-binding MEEVD motif at its extreme C-terminus.

The much more elaborate structure of RPAP3 in human R2TP allows for a far greater level of adaptability, while retaining the topological proximity of TPR and PIH domains required to bring HSP90 and client together. RPAP3 is divided into two regions that are located at the two opposite faces of the RUVBL1–RUVBL2 ring. As revealed in our cryo-EM structures, the interaction of the RBD of RPAP3 with the ATPase face of the RUVBL1–RUVBL2 ring provides a tight anchor for the C-terminus of the protein, while allowing considerable flexibility for the CS-binding segment, TPR domains and N-terminus of the protein on the other face, coupled to the RBD by the long flexible central segment that spans the rim of the ring.

As the necessary and sufficient interaction of the C-terminal α-helical bundle RBD of RPAP3 occurs with a surface of RUVBL2 presented on the 'uncluttered' ATPase face, the heterohexameric RUVBL ring is capable of binding up to 3 RPAP3 molecules, and the presence of 3 symmetrically equivalent RBDs is evident in cryo-EM images of saturated complexes. Nonetheless, the number of RPAP3 molecules per RUVBL ring in vivo, and/or in the context of a larger assembly, cannot be determined by our work. Cryo-EM shows that PIH1D1, with which RPAP3 also interacts, binds asymmetrically to the DII domain-face of the ring in a similar location as in the yeast R2TP complex[24,25]. The RPAP3–PIH1D1 sub-complex behaved as an elongated heterodimer in sedimentation velocity experiment, with an estimated average molecular mass of $103,900 \pm 320$ Da determined by sedimentation equilibrium assays (Supplementary Fig. 6), which corresponds to a 1 RPAP3:1 PIH1D1 heterodimer (104,212 Da). We have been unable to define PIH1D1 within R2TP at high resolution, due to the flexibility in this region. However, based on the dimensions of this region in the maps, we speculate that it could be conceivable that the RUVBL ring may only accommodate 1 PIH1D1 molecule, as in yeast[24,25]. In this hypothetical scenario, while three RPAP3 molecules may simultaneously bind to the three RUVBL2 subunits in the AAA-ring, only one at a time could be fully engaged through the additional interaction with the single copy of PIH1D1. This could provide a mechanism whereby multiple copies of HSP90, bound to the TPR domains, could be brought to bear, or additional client components potentially recruited via the poorly understood N-terminal region of RPAP3, could be introduced—further work will be required to test these possibilities.

The R2TP core complex analyzed here was reconstituted in vitro, without subunits of the associated PFDL module or other interacting proteins whose presence would add still further complexity. These additional components may facilitate recruitment of specific clients, such as RNA Pol II or PIKKs, but may also have important functional influence on the conformational state of the R2TP core components. It is also likely that there will be steric competition or collaboration between some of these additional components. For instance, the ZNHIT2 protein was recently found to bind RUVBL2 and it has been proposed to bridge R2TP and U5 snRNP[5]. The structure of the RUVBL1–RUVBL2–RBD complex we describe here raises the possibility that RPAP3 and ZNHIT2 could either compete or collaborate for binding to RUVBL2. Since each RUVBL1–RUVBL2 ring contains 3 RUVBL2 molecules, complexes containing RPAP3 and ZNHIT2 proteins could be also conceptually possible.

Along with the structure of the yeast R2TP complex[24,25], the cryo-EM structure of the human R2TP core presented here provides a clear understanding of the architecture and evolution of this complex HSP90 co-chaperone. These studies suggest a mechanism for how R2TP brings HSP90 and clients (or client

adaptors) into proximity, and at least for yeast suggest some involvement of the ATPase activity of the RUVBL proteins in modulating this, although the significance of this for R2TP function in vivo is far from clear. The ultimate question of how HSP90 functions with R2TP/PFLD to facilitate the assembly of large multiprotein complexes such as Pol II remains to be determined.

## Methods

**Cloning**. N-terminal His-tagged *RuvBL1* and untagged *RuvBL2* were cloned as indicated in Lopez-Perrote et al.[28]. For pull-down experiments, a 3xMyc tag was incorporated to the N-terminus of RUVBL1. The *RPAP3* full length (FL) was purchased from GenScript. The *RPAP3₁₋₄₃₀*, *RPAP3₁₋₄₀₀*, *RPA3₁₋₄₂₀*, *RPAP3₃₉₅₋₆₆₅*, *RPAP3₄₃₀₋₆₆₅*, *RPAP3₅₂₃₋₆₆₅*, *RPAP3₄₃₀₋₅₄₁*, *RPAP3₅₄₁₋₆₆₅*, and *RPAP3₃₉₅₋₆₆₅* (RPAP3-ΔNT) gene truncations were cloned using NdeI sites and ligation-free cloning using infusion cloning (Clonetech lab Inc.) into a modified pGex6p modified plasmid named p3E (University of Sussex, UK), which resulted in N-terminal GST-tagged *RPAP3* genes (Supplementary Table 1).

The human *PIH1D1* full-length gene and *PIH1D1₁₈₀₋₂₉₀* truncated genes were cloned into a pET28b using NdeI site, which resulted in 6XHis-hPIH1D1. PIH1D1 constructs were co-expressed with RPAP3 since PIH1D1 was insoluble when expressed alone. RuvBL1 full-length gene was cloned into NheI and BamHI sites of a modified pRSETA plasmid (containing 3×Myc tags) and *RuvBL2* gene was cloned into pET28b using NdeI and BamHI sites. Human *Hsp90* full-length beta gene was cloned into modified pET28b plasmid (contacting 6His-2Xstrep-tags) using NdeI, which resulted in 6His-2Xstrep-PreSc-Hsp90FL beta gene.

**Protein expression and purification**. The human RUVBL1–RUVBL2 protein complex and RUVBL2 used in the pull-down experiments, and human HSP90 beta were transformed into *Rosetta (DE3) pLysS* cells (F⁻*ompT hsd*S_B(r_B⁻ m_B⁻) *gal dcm* (DE3) pLysSRARE (Cam^R), Merck Millipore Ltd.). The cells were grown in the presence of ampicillin and kanamycin antibiotics at 37 °C until the cells reached their log-phase. Then the cells were induced by the addition of 1 mM IPTG. The cells were further grown at 25 °C overnight for protein expression. The cell mass was pelleted by spinning the cell culture at 6238×*g* for 10 min. The cells were lysed using a sonicator in 20 mM HEPES pH 7.5, 140 mM NaCl (HEPES buffer) and 1 tablet of EDTA-free protease inhibitor (Sigma-Aldrich Ltd.). The cell lysate was centrifuged at 20,000×*g* for an hour at 4 °C. The clear supernatant was loaded on to the equilibrated Talon beads in 20 mM HEPES pH 7.5, 140 mM NaCl for His-tag affinity-chromatography. The beads were washed with HEPES buffer to remove the contaminant proteins. The proteins of interest were eluted with 500 mM imidazole in HEPES buffer. The eluted proteins were analyzed by 4–12% SDS-PAGE and concentrated to 6 mg ml⁻¹ using viva-spin (MWCO 10K, Sartorius). The proteins were further purified by size-exclusion chromatography (SEC) using S200 10/300 column (GE Healthcare Ltd.). The GST-tagged RPAP3FL and RPAP3 shorter gene constructs alone and in complex with PIH1D1 full length and PIH1D1₁₈₀₋₂₉₀ were expressed in *Rosetta (DE3) pLysS* cells. The cells were lysed similar to RUVBL1/2 and HSP90 beta. The proteins were purified using GST-tag affinity chromatography by adding the clear cell lysate to the GST-beads. The GST-bound proteins were washed with HEPES buffer and they were eluted with 50 mM glutathione. The proteins were incubated with PreScission protease (3C protease) to cleave the GST-tag at 4 °C overnight and the proteins free of GST-tag were concentrated to 20 mg ml⁻¹ using viva-spin with MWCO of 10k. The proteins were further purified by gel filtration chromatography using S200 26/60 column in degassed 20 mM HEPES pH 7.5, 500 mM NaCl. For those pull-down experiments using untagged RUVBL1, 6xHis-tagged RUVBL1 was purified and the tag cleaved using PreScission protease.

RUVBL1–RUVBL2 complexes used for cryo-EM studies were produced and purified as before[28]. N-terminal His₁₀-tagged human RuvBL1 and untagged human RuvBL2 were co-transformed into *Escherichia coli* BL21 (DE3) cells grown in LB medium. The lysate was applied to a HisTrap HP column (GE Healthcare) equilibrated in 50 mM Tris-HCl pH 7.4, 300 mM NaCl, 10% (v/v) glycerol, and 20 mM imidazole. Elution was performed using a 20–500 mM imidazole gradient, followed by a preparative SEC using a Sephacryl S300 column (GE Healthcare) equilibrated in 50 mM Tris, 300 mM NaCl and 1 mM DTT. The uncropped SDS-PAGE gels of the SEC of the RUVBL1–RUVBL2 and RPAP3–PIH1D1 used for cryo-EM are shown in Supplementary Fig. 7. The purification was monitored by SDS–PAGE. RUVBL1 used for the pull-down experiments contained an N-terminal 3xMyc tag, to help distinguish RUVBL1 and RUVBL2 in SDS-PAGE.

**Pull-down assay for interaction mapping**. Twenty micromolar GST-RPAP3 (full length and different lengths of RPAP3) and RPAP3–PIH1D1 and RPAP3–PIH1D1₁₈₀₋₂₉₀ complexes were mixed with 30 µl GST beads, which were equilibrated in 50 mM HEPES pH 7.5, 140 mM NaCl. Sixty micromolar of the RUVBL2 alone and RUVBL1–RUVBL2 complex were added to the above mixture for the interaction mapping study. In these experiments, a 3xMyc tag was incorporated to the N-terminus of RUVBL1 to help in distinguishing RUVBL1 and RUVBL2 in SDS-PAGE. The above reaction mixture was incubated for 45 min at

4 °C rotating at 20 rpm min⁻¹. The beads were washed three times with 500 µl of HEPES buffer and the bound fraction was eluted with 50 mM glutathione.

Human R2TP complex assembly was monitored by pull-down. For this, the purified RPAP3–PIH1D1 and RUVBL1–RUVBL2 proteins were used for the R2TP complex assembly. The 20 µM GST–RPAP3–PIH1D1 complex and 60 µM RUVBL1–RUVBL2 proteins were used in the experiments; 30 µl GST-beads (GE Healthcare) equilibrated in 50 mM HEPES pH 7.5, 140 mM NaCl (HEPES buffer) were added to the above protein complexes. Then the protein mixture was incubated for 45 min rotating at 20 rpm at 4 °C. The beads were washed three times with HEPES buffer to remove the non-specific proteins bound on to the GST-beads. The bound fraction was eluted with 50 mM glutathione in 20 mM HEPES, pH 7.5, 140 mM NaCl. For the co-expression experiments shown in Fig. 1, we co-expressed Gst-RPAP3FL, Gst-RPAP3FL-RuvBL2FL, Gst-RPAP3₅₄₁₋₆₆₅-RuvBL2FL and Gst-RPAP3₅₄₁₋₆₆₅-RuvBL2 ΔDII. The interactions were analyzed using GST-affinity chromatography and the bound fractions were eluted with 50 mM glutathione in the 20 mM HEPES pH 7.85, 140 mM NaCl, and the quality of the protein complex was visualized using SDS-PAGE.

**Cryo-EM of human R2TP**. To image R2TP at high resolution, 0.45 µM RUVBL1–RUVBL2 (estimated as dodecamers) was incubated with 9 µM of RPAP3–PIH1D1 complex for 20 min in ice and the mix dialyzed for 5 h against 25 mM HEPES pH 7.8, 130 mM NaCl, 10 mM 2-Mercaptoethanol. After dialysis, the sample was recovered and incubated with ADP (pH 7.0) for 1 h at a final concentration of 0.5 mM. Subsequently, aliquots of 2.5 µl were applied to Quantifoil R1.2/1.3 carbon grids after glow-discharge, and then flash frozen in liquid ethane. An initial test data set was obtained in Grenoble Instruct Center, France, using a FEI Polara microscope operated at 300 kV. High-resolution structures were obtained from data collected in a Titan Krios (eBIC, Diamon, Oxford, UK), automatically, with three images per hole, using a GATAN K2-Summit detector in counting mode and a slit width of 20 eV on a GIF-Quantum energy filter (Supplementary Tables 2, 3).

**High-resolution image processing of RUVBL1–RUVBL2–RBD**. As general methodology, MotionCorr2[43] was used for whole-frame motion correction, GCTF[44] for estimation of the contrast transfer function parameters, and RELION-2.0[45] for all subsequent steps. Local motion was corrected in MotionCorr2 dividing the frame in 36 patches (6 × 6 patches), with dose weighting. A manually picked subset of micrographs was used to obtain 2D references for template-based particle picking. The selected particles were then submitted to several rounds of 2D and 3D classifications, to discard low-quality particles and some remaining RUVBL1–RUVBL2 dodecamers. Low-pass filtered versions of previous structures were used as starting point of classifications and refinements, to reduce bias.

A specific classification protocol was designed to analyze the stoichiometry of RBDs bound to the RUVBL ring. For this, Class3D in RELION was used but using a mask covering the edge of the RUVBL ring and the regions outside the ring. Particles were split in up to eight groups using this focused classification strategy. The majority of particles were grouped into one class, containing 3 RBDs and corresponding to 67.3% of the particles. The remaining classes corresponded to particles containing 2 RBDs (16.4%) or 1 RBD (16.3%). Images of R2TP containing 3 RBDs were then processed applying 3-fold rotational symmetry and using standard procedures in RELION[45].

For the refinement of the RUVBL1–RUVBL2–RBD complex at high resolution, 96,406 particles showing the best parameters after 2D classification in RELION were selected and subjected to a round of automatic 3D refinement in RELION to generate a consensus 3D model. When refinement was performed without applying rotational symmetry, similar results were obtained, but some differences in the quality of the different RBDs bound to one RUVBL ring, suggested that the complex had a rigid conformation at the RUVBL ring, and relatively flexible RBDs (or variable quality). To improve the quality of the structure defining the interaction between the RBD and the RUVBL ring, we applied the method previously developed to solve a similar situation for the structure of the apoptosome[32]. For this, each particle was rotated 120 and 240° so that all RBDs were then placed in the same position. Subsequently, particles were classified using the Class3D utility in RELION and using a mask representing 1 RBD and the ring of RUVBL1–RUVBL2. The most populated class was automatically refined using the Ref3D utility using the same mask used for Class3D, local search of angles and without applying symmetry. Further details on the strategy applied here can be found in the Methods section of Zhou et al.[32]. When applied to our data, resolution was improved from 3.8 Å when applying 3-fold rotational symmetry to 3.57 Å, estimated using gold standard Fourier Shell Correlation (FSC) between two independent maps using cut-off of FSC = 0.143. B factor sharpening was performed using automatic procedures in Relion2. Local resolution was estimated using ResMap[46]. Structures were visualized using UCSF Chimera[47].

**Cryo-EM and processing of RUVBL1–RUVBL2–RBD and R2TP-ΔNT**. The reconstitution of complexes between RUVBL1–RUVBL2 and RPAP3₄₃₀₋₆₆₅, and the reconstitution of R2TP using the RPAP3–ΔNT–PIH1D1 sub-complex instead of RPAP3–PIH1D1 were performed with the same protocol used for the R2TP used in cryo-EM experiments. For consistency, all these observations were performed

with the same buffer conditions used for the assembly of R2TP, and in the presence of ADP. Vitrifications, the general image processing, 2D classifications, and the generation of 2D averages and 3D volumes were performed following similar strategies to those described for the R2TP images, but the cryo-EM micrographs were collected in a 200-kV FEI Talos Arctica operated with a FEI Falcon II detector and located at the Centro Nacional de Biotecnologia (CNB) in Madrid.

**Image processing of flexible regions**. For refinement of the flexible regions of the R2TP complex, the same set of 96,406 particles used to resolve the structure of RUVBL1–RUVBL2–RBD was classified searching for a subset of particles of R2TP with a more homogenous conformation for the flexible regions after extensive 3D classification steps using Relion. This subset, containing a selection of 27,385 particles, was then refined, converging to a structure with an estimated average resolution of 8.72 Å using gold standard Fourier Shell and FSC = 0.143.

To analyze the structural details for the interaction between PIH1D1 and RUVBL1–RUVBL2, the images of R2TP complexes containing 3 RBDs were refined without symmetry and using a mask that removed the influence of the flexible regions except the vicinities of the DII domains in RUVBL1–RUVBL2. One subset of 182,351 particles corresponding to complexes containing 3 RBDs and that displayed a defined density in DII-domain face of the RUVBL1–RUVBL2 reached an estimated resolution of 6.58 Å using gold standard Fourier Shell and FSC = 0.143.

**Modeling and refinement of RUVBL1–RUVBL2–RBD**. De novo modeling of the C-terminal domain of RPAP3 was performed using a strategy based on homology modeling and molecular dynamics simulation. First, analysis of the sequence in the secondary structure prediction server PSIPRED revealed 8 α-helices consecutively, starting from Ala547 to C-terminus Gly665. A long and disordered region of RPAP3 N-terminal to this helical domain contributed to a clear identification of the domain. The sequence of RPAP3-RBD domain (residues 541–665) was submitted to I-TASSER homology modeling server[35], which provided up to five different atomic models for the query sequence. The best solution according to the I-TASSER scoring was fitted within the target cryo-EM map.

The target cryo-EM map density contained 8 α-helices, the same as the predicted atomic model in I-TASSER. The prediction was first fitted as a rigid body in the map, followed by a flexible fitting using molecular dynamics (MD) simulations in AMBER[48] (Supplementary Movie 1). Flexible fitting was performed in two orientations: one from the initial fitted conformation and the second forcing the reverse orientation as control. The reverse orientation was discarded since no reasonable fitting of the model into the cryo-EM map was possible after the simulation.

A model for RUVBL1–RUVBL2–RBD was built initially by the fitting as a rigid body of the crystal structure of the human RUVBL1–RUVBL2 truncated hexamer (PDB 2XSZ)[29] into the cryo-EM map with the addition of the model generated for the RBD using AMBER. The full structure of the RUVBL1–RUVBL2–RBD was refined using Phenix[49] and Coot[36].

The information of the crystal structure of the RPAP3 fragment comprising residues 578–624 helped to model the connectivity between helix 5 and helix 6 of the RBD, which faces RUVBL2. Density for side chains was also visible in most of the helices of the RBD (Supplementary Fig. 5), and this information was used during modeling. Detailed protein–protein interactions are mostly discussed at the level of secondary structural elements, except where side chain density was clear.

**R2TP reconstitution and purification using GraFix**. For the experiments using GraFix[50], the different complexes were analyzed using a linear 10–40% sucrose gradient together with a 0–0.15% of glutaraldehyde gradient. Fifty microliters of the mixture were used for each gradient and run at 125812 g using a SW60Ti rotor, 16 hours and 4 °C. Fractions of 100 μl were collected from top to bottom of the gradient, the fixation reaction was stopped by adding glycine pH 7.0 at final concentration of 100 mM. Blue-Native system (Invitrogen) was used to analyze the fractions.

**Cross-linking coupled to mass spectrometry**. R2TP was reconstituted as for the cryo-EM experiments and cross-linked with isotopically coded N-hydroxysuccinimide (NHS) esters disuccidinimidyl suberate (DSS $H_{12}/D_{12}$) and bis-sulfosuccidinimidyl suberate (BS3 $H_{12}/D_{12}$) (Creative Molecules, Canada) at a final excess concentrations of 100 and 250×. The reactions were incubated for 45 min at 37 °C, and quenched by adding 50 mM $NH_4HCO_3$ (final concentration) for another 15 min.

The cross-linked sample was freeze-dried and then resuspended in 50 mM $NH_4HCO_3$ to reach 1 mg ml$^{-1}$ final protein concentration. The sample was then reduced using 10 mM DTT and alkylated with 50 mM iodoacetamide. Subsequently, proteins were digested with trypsin (Promega, UK) using 1:20 enzyme-to-substrate ratio, at 37 °C, and the incubation was done overnight. A final concentration of 2% (v/v) formic acid was added to acidify the samples, and the peptides were fractionated by peptide SEC in a Superdex Peptide 3.2/300 column (GE Healthcare) with 30% (v/v) acetonitrile/0.1% (v/v) TFA as mobile phase and using a flow rate of 50 μl min$^{-1}$. Fractions were collected, lyophilized, and resuspended in 2% (v/v) acetonitrile and 2% (v/v) formic acid.

Fractions were analyzed by nano-scale capillary LC–MS/MS using an Ultimate U3000 HPLC (ThermoScientific Dionex, USA) and a flow of approximately 300 nl min$^{-1}$. Peptides were separated on a C18 Acclaim PepMap100 3 μm, 75 μm × 250 mm nanoViper (ThermoScientific Dionex, USA) and eluted with a acetonitrile gradient. The analytical column outlet was directly interfaced via a nano-flow electrospray ionization source, with a hybrid dual pressure linear ion trap mass spectrometer (Orbitrap Velos, ThermoScientific, USA). A resolution of 30,000 was used for data-dependent analysis for the full mass spectrometry spectrum, followed by 10 MS/MS spectra in the linear ion trap. Mass spectrometry spectra were collected over a 300–2000 m/z range. MS/MS scans were collected using threshold energy of 35 for collision-induced dissociation.

For data analysis, Xcalibur raw files were converted into the open mzXML format through MSConvert (Proteowizard) with a 32-bit precision, and the converted files were directly used as input for xQuest searches on a local installation (http://prottools.ethz.ch/orinner/public/htdocs/xquest/). The following criteria were used for the selection of cross-linked precursor MS/MS data: a mass shift of 12.07532 Da among the heavy and the light cross-linkers; precursor charge ranging from 3+ to 8+; maximum retention time difference 2.5 min. Searches were performed against an ad hoc database containing the RUVBL1, RUVBL2, PIH1D1, and RPAP3 sequences plus their reverse as decoy. A number of parameters were set to perform the xQuest searches: maximum number of missed cleavages (excluding the cross-linking site) 3; peptide length 4–50 amino acids; fixed modifications carbamidomethyl-Cys (mass shift 57.02146 Da); mass shift of the light cross-linker 138.06808 Da; mass shift of mono-links 156.0786 and 155.0964 Da; MS1 tolerance 10 ppm, MS2 tolerance 0.2 Da for common ions and 0.3 for cross-link ions; search in enumeration mode (exhaustive search). The following criteria were used to filter search results: MS1 mass tolerance window −3 to 7 ppm. Finally, each MS/MS spectra was manually inspected and validated.

**Sedimentation velocity assay of TP complex**. Four-hundred microliters of 10 μM and 5 μM RPAP3–PIH1D1 prepared in 25 mM Hepes, 130 mM NaCl, 0.1 mM TCEP, pH 7.8, were loaded into analytical ultracentrifugation cells. The experiments were carried out at 10 °C and 149,103×g in an XL-I analytical ultracentrifuge (Beckman-Coulter Inc.). This was equipped with UV-VIS absorbance and Raleigh interference detection systems, and the sedimentation profiles were recorded at 280 nm. Least-squares boundary modeling of sedimentation velocity data and the continuous distribution c(s) Lamm equation model was used to calculate sedimentation coefficient distributions, as implemented by SEDFIT 14.1[51]. The program SEDNTERP[52] was used to correct experimental s values to standard conditions (water, 20 °C, and infinite dilution) to obtain the corresponding standard s values (s20,w).

**Sedimentation equilibrium assay**. Short column (90 μl) Sedimentation Equilibrium experiments were carried out at speeds ranging from 4536×g to 6532×g and at 286 nm, and using the same experimental conditions as those described for the Sedimentation Velocity experiments. A high-speed centrifugation run using 185,795×g was performed to estimate the corresponding baseline offsets after the last equilibrium scan. Weight-average buoyant molecular weights of proteins were determined by fitting a single species model to the experimental data using the HeteroAnalysis program, and corrected for solvent composition and temperature using the program SEDNTERP[52].

**RPAP3$_{523-665}$ expression, purification, and crystallization**. The gene encoding RPAP3$_{523-665}$ fragment was cloned into the NdeI and BamHI site of the p3E plasmid (home grown plasmid from University of Sussex). BL21 (DE3) E. coli cells (F⁻ ompT gal dcm lon hsdSB(rB- mB-) λ(DE3 *lacI lacUV5-T7gene 1 ind1 sam7 nin5), NZYTech) were transformed with p3E plasmid containing RPAP3$_{523-665}$. The transformed cells were grown in LB media at 37 °C for 5 h followed by induction with 1 mM IPTG. The cells were further grown for 15 h at 20 °C. Cells were harvested using 5000×g for 10 min. The cell pellet was re-suspended in 20 mM HEPES pH 7.8, 140 mM NaCl, 0.5 mM TCEP, and sonicated at 4 °C. The cell lysate was spun at 20,000 g for 1 h at 4 °C and the supernatant was used for the GST affinity chromatography. The purified GST-RPAP3$_{523-665}$ protein was treated with PreScission protease overnight at 4 °C to remove the GST-tag and the protein was further purified by SEC.

Purified RPAP3$_{523-665}$ protein was concentrated to 8 mg ml$^{-1}$ using 10k MWCO viva-spin (Sartorius). The crystallization trials were set up using 0.2 μl protein and 0.2 μl crystallization screen buffer using sitting drop method. The crystallization trial trays were incubated at 14 °C. Small crystals appeared in 0.5 M potassium thiocyanate, 0.1 M bis-tris propane pH 7.0 (well H6) of SaltRX crystal screen (Hampton research Ltd) after 1 month of incubation. The crystals were flash frozen in liquid nitrogen using 30% glycerol as a cryo-protectant and the data were collected at Diamond Light source, UK. Wavelength used was 0.9763 Å.

The data were processed using standard methodology, and programs of the CCP4 suite[53], Xia2[54], REFMAC[55], BUSTER (https://sbgrid.org/software/titles/buster), and COOT[36], together with the ARCIMBOLDO software[56] (Supplementary Table 4).

**Data availability**. Data supporting the findings of this manuscript are available from the corresponding authors upon reasonable request.

The EM maps have been deposited in EMDB and PDB: EMD-4289 (R2TP-C3symmetry); EMD-4287 (R2TP-1RBD); EMD-4290 (R2TP-subgroup1); EMD-4291 (R2TP-subgroup2); PDB 6FO1 (model of RUVBL1–RUVBL2–RBD with 1RBD), and PDB 6FM8 (crystal structure of RBD fragment).

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

## Acknowledgements

This work was supported by the Spanish Ministry of Economy, Industry and Competitiveness and the "Agencia Estatal de Investigación" (MINECO/AEI), co-funded by the European Regional Development Fund (ERDF) (SAF2014-52301-R and SAF2017-82632-P to O.L., SAF2014-59993-JIN to F.M., and BES-2015-071348 to C.F.R.), the Spanish National Research Council (i-LINK0997 to O.L.), a Wellcome Trust Senior Investigator award (095605/Z/11/Z), and Award Enhancement Grant (095605/Z/11/A) (to L.H.P.). We acknowledge Diamond for access and support of the Cryo-EM facilities at the UK national electron bio-imaging centre (eBIC), proposals EM13312, EM13520, and EM15997, funded by the Wellcome Trust, MRC and BBSRC. Preliminary work used the platforms of the Grenoble Instruct-ERIC Center (ISBG: UMS 3518 CNRS-CEA-UGA-EMBL) with support from FRISBI (ANR-10-INSB-05-02) and GRAL (ANR-10-LABX-49-01) within the Grenoble Partnership for Structural Biology (PSB). The electron microscope facility is supported by the Rhône-Alpes Region, the Fondation Recherche Medicale (FRM), the funds FEDER, the CEA, and the GIS-Infrastrutures en Biologie Sante et Agronomie (IBISA). We also acknowledge the help of Guy Schoehn (IBS-Grenoble). INSTRUCT and iNEXT supported access to the cryo-EM facilities. We thank the EM Units of the CIB-CSIC and the CNIO for support in preparing and screening the grids.

## Author contributions

L.H.P. and O.L. outlined the project and supervised the research. L.H.P. and M.P. designed the pull-down experiments. O.L., F.M., and H.M.-H. designed the cryo-EM experiments. M.P. expressed and purified TP, and performed the pull-down experiments for the mapping of interactions. H.M.-H. and F.M. purified RUVBL1–RUVBL2. H.M.-H. and F.M. assembled the R2TP complex for GraFix, negative stain and cryo-EM. F.M. and H.M.-H. collected cryo-EM data, and R.N. and D.G.-C. helped to prepare grids. F.M. and C.F.R. solved the high-resolution cryo-EM map of the RUVBL1–RUVBL2–RBD complex, performed and analyzed de novo modeling of the RBD domain. H.M.-H. solved/obtained the maps of PIH1D1 in the complex and performed the structural study of the flexible regions. H.M.-H. assembled the complexes, performed cryo-EM, and collected and processed the data for all the mapping experiments using GST-RPAP3 and isolated RBDs bound to RUVBL1–RUVBL2. G.D. and M.S. performed XL-MS. M.P. and S.M.R. performed the crystallographic studies. C.P. helped analyze and interpret the results. L.H.P. and O.L. wrote the manuscript.

## Additional information

**Competing interests:** The authors declare no competing interests.

