## [Peer Review File · Nature Communications]

Reviewers' comments:

Reviewer #1 (Remarks to the Author):

In this manuscript Lorca and colleagues provide interesting data on the structure of the R2TP cochaperone and how human R2TP differs from its yeast counterpart. Whereas much progress has been made in recent years in elucidating the structure of *S. cerevisiae* R2TP, the human complex has remained poorly studied. While potentially interesting, this manuscript suffers from one major caveat: it acknowledges that RPAP3 is structurally different from yeast Tah1p, but falls short of applying that same logic to human R2TP as a whole by neglecting to consider (and even mention) any of the other subunits than the four that make up the yeast complex. Indeed, while there is much evidence showing that human R2TP also contains the prefoldin and prefoldin-like proteins PFDN2, PFDN6, URI1, UXT, PDRG1 (which is why it is usually referred to as R2TP/prefoldin-like or R2TP/PFDL) as well as RNA polymerase subunit POLR2E/RPB5 and WDR92/Monad, not one article published so far has conclusively demonstrated that human R2TP can exist solely as a combination of RUVBL1, RUVBL2, RPAP3 and PIH1D1 as is the case in *S. cerevisiae*.

Because all proteins used in this article were produced ectopically in bacteria and because the R2TP complex was spontaneously assembled by mixing together a subset of its subunits *in vitro*, the potential for artefactual interaction is high. The best procedure would have been to purify the complex directly from human cells and proceed with structural analysis. However, it is significantly more difficult to purify a fully-assembled protein complex from a cell lysate with enough purity to perform cryo-EM, which is likely why the authors opted for the recombinant protein shortcut. It is important that interpretation of the observations derived from this approach is tempered and presented as hypothesis rather than conclusions.

Moreover, the C-terminal moiety of RPAP3, claimed here to be "novel", has been reported previously and is already annotated as a bona fide protein domain (pfam13877) that is present in two other human proteins: CCDC103 and SPAG1. It is unfortunate that the manuscript failed to recognize this fact since interaction data for these two proteins tend to corroborate the hypothesis that this moiety is indeed involved in binding RUVBL2, as CCDC103 was shown to copurify with RUVBL2 (PMID: 25416956), while SPAG1 was observed with the RUVBL1-RUVBL2 interactor C12orf45 (PMID: 27173435).

Lack of thorough review of the literature published so far on R2TP/PFDL and its cofactors can also be surmised by the authors claim that a number of biological aspects pertaining to the cochaperone are still poorly understood. For example, the manuscript states that "how RNA Pol II subunits are recruited to R2TP and how R2TP and HSP90 contributes to Pol II assembly is currently unknown" (line 62). While not entirely wrong, it is a rather bold statement as URI1, one of the R2TP/PFDL subunits disregarded by the authors, is a well-known direct interactor of RPB5 (a RNA Pol shared subunit) that was shown to affect assembly of all three nuclear RNA polymerases. The experiments were performed on the yeast homolog of URI1, Bud27, while no such activity as yet to be reported for yeast R2TP, underlining the paramount importance of considering the entirety of the human R2TP/PFDL complex rather than focusing on its yeast-like R2TP module. The manuscript also states that "neither Pol II nor snoRNPs subunits contain [the PIH-binding] motif, and must therefore be recruited to R2TP through alternative mechanisms, yet to be described" (line 71). While PIH-binding motifs are indeed required for PIKK interaction through Tel2, binding to snoRNPs is mediated by NUFIP1 and ZnF-HIT domain proteins ZNHIT3 and ZNHIT6. Additionally, interaction to the most recently discovered client complex of R2TP, U5 snRNP (regrettably not mentioned in this manuscript) appears to be mediated by a conjunction of both a PIH-binding motif protein (ECD) and a ZnF-HIT protein (ZNHIT2). Given that ZnF-HIT proteins were reported to interact with RUVBL2, it would be interesting for the authors to address that fact and raise the possibility of there being a competitive or collaborative interaction of RPAP3's C-terminal domain with this protein family.

In sum, the present manuscript would benefit from 1) a serious effort to better review the literature, 2) the authors being more careful with claims of primacy in discovering RPAP3's C-terminal domain, 3) addressing the fact that subunits are missing from the assembled complex and a more nuanced and conservative interpretation of the resulting structural data. This being said, the research presented in this manuscript is interesting enough to warrant publication in *Nat Commun* provided that 1) my comments above are taken into account in a revised manuscript, and 2) the manuscript is being reviewed by a protein structural biologist who might better look over the experimental methodology employed here (not being myself a specialist in structural studies). Understanding the structure of human R2TP is likely to accelerate research on this "most complex HSP90 cochaperone yet described".

Minor points:

1) Line 54-55: The first demonstration of an interaction with Pol II subunits comes from Jeronimo 2007. This same paper proposed the name RPAP3 for the first time (RNA Pol II Associated Protein 3). The reference Cloutier 2009 describes for the first time the 11-subunit R2TP/PFDL complex. Please clarify and add missing ref.

2) Line 168: I suggest continue using RUVBL1-RUVBL2-RBD all along the manuscript (instead of RBD-RUVBL1-RUVBL2).

Reviewer #2 (Remarks to the Author):

RPAP3 provides a flexible scaffold for coupling HSP90 to the human R2TP co-chaperone complex. Fabrizio Martino et al.

The authors have done a series of experiments including cryo-EM and cross-linking mass spectrometry, which suggest that RPAP3 interacts within R2TP complex directly with ATPase domain of RUVBL2 through its RUVBL-binding-domain (RBD) located in the RPAP3 C-terminus. They also show that (in contrary to the small Tah1p molecule present in yeasts instead of RPAP3) RPAP3 spans to the opposite face of the RUVBL single ring and provides a platform for bringing HSP90 and R2TP client proteins together. These novel findings uncover the architecture of R2TP complex and elucidate how the R2TP complex brings its substrate to the proximity of HSP90. Overall, the manuscript is well written and the data presented are of high quality.

I have only one comment - the authors use untagged RUVBL2 and RUVBL1 tagged with 3xmyc tag. However, the method part of the paper doesn't state whether the tag is C or N terminal and whether it could compromise RUVBL1 interaction with RPAP3. Have the authors tried to do a similar experiment to the one in Figure 1d with RUVBL1 untagged and RUVBL2 tagged with 3xmyc and would they be able to get the same results as shown in Figure 1D?

Reviewer #3 (Remarks to the Author):

Martino et al. describe structural characterization of human R2TP complex. The group previously solved a low resolution structure of the yeast R2TP by cryoEM in which the Pih1-Tah1 was found to decorate the top face of the Ruvb1/2 heterohexameric ring. In this study, the authors reconstituted human R2TP and determined a cryoEM structure. The key difference between human and yeast R2TP is in the RPAP3/Tah1 subunit. The human homolog of yeast Tah1, RPAP3, is three times of Tah1 in size. A C-terminal domain of RPAP3, namely residues 541-665 (RBD), is found to associate with RUVBL2, not RUVBL1. Otherwise, human R2TP resembles its yeast counterpart and

the TP components similarly bind to domain II of the AAA+ ATPases. RPAP3 also uses a short peptide (400-420) to interact with the CS domain of PIH1D1. The novel aspect of this study is the discovery of the binding mode of the accessory RBD domain that provides additional anchor for the TP molecules on the ATPases. However, both structural and biochemical are preliminary and not sufficient to validate the proposed assembly model. Authors additionally characterized Hsp90 binding to R2TP biochemically but provided no structural follow-up. The major concerns are:

1) Authors seem to take it granted that human R2TP forms a hexamer. They stated that TP binding disrupted the R2 dodecamers but with minimal evidence. This is an important point, especially authors also state that RBD-RUVBL2 interaction is the “primary” interaction between R2 and TP. If RBD binds the ATPase domain of R2, it may well attach to the dodecamer without disruption of the oligomeric state. In a related concern 3), authors added ADP to the incubated R2TP, which is known to impact R2 oligomerization. Thus, it is not convincing at all that TP disrupts the dodecamer and the nature of human R2TP assembly remains unresolved;

2) Despite cryoEM criteria suggest that the RBD-R2 structure is at a 3.6Å overall resolution, many problems are associated with this structural model. The first is about the binding stoichiometry. Classes of cryoEM particles seem to show a distribution of 1, 2 or 3 RBD bound with R2. However, authors had to play a trick in order to increase signal-to-noise ratio by rotating all RBD into one RUVBL2 subunit such that the structure appears to be 1 RBD to R2 hexamer and this is what depicted in the final summary model (Fig. 7). Obviously, this is incorrect. Authors must make efforts in characterizing the stoichiometry by an independent method, as this property is very relevant to human R2TP structural model. The second concern is about RBD structure itself. While it is clear that the region 541-665 binds R2, the exact interface between RBD and R2 is unambiguous. As authors maybe well aware of the limit of protein threading models, the actual registering of amino acids used to interpret the density of the helices may be incorrect. Authors did not provide any images of how side chains of RBD residues fit the density nor did they provide correlation coefficients for readers to judge the quality of the model. Biochemically, key residues at the interface should be disrupted and their effects on binding assessed.

3) Authors used ADP during reconstitution without providing any reasoning. What is the impact of ADP on R2TP assembly in their hands? Was this simply done to increase homogeneity? Which functional state of the ADP-bound human R2TP is?

4) Structural characterization of TP binding to DII seems to be very difficult, which adds further confusion in stoichiometry. Authors presented two maps – one at 8.72 Å by classification and refinement of the entire asymmetric particles and another at 9.48Å resulted from a masked region consists of DII and PIH1D1. The first map suggests that 3 RPAP3 are associated (Fig. 6b) whereas the second map suggests a single PIH1D1 is bound (Fig. 6a) to R2. On the other hand, biochemical data seem to suggest PIH1D1:RPAP3 is 1:1. How should these discrepancies be resolved? Curiously, there are no cross-links detected between PIH1D1 and any other components, which could result if PIH1D1 is primarily detached from DII. These issues seem to suggest problems either in sample preparation or structural characterization.

Other related concerns:

λ The titles in Supplementary Table II are very confusing. One refinement was assigned to R2-RBD-PIH1D1 and the other to R2TP subpopulation. If I understand it correctly, it is the same data but refined differently. Please find a way to clarify these different reconstructions.;

λ Some example density maps are provided in Fig. S2 but they appear to be the well conserved regions within R2 rather than regions within RBD. It would be important to validate the tracing of RBD;

λ Figure 1b. All constructs shown in this figure resulted in solubilization of PIH1D1. Did authors

have examples of constructs that did not solubilize PIH1D1 that can provide a support for the proposed role of 400-420 (for instance, GST only)? What are the upper bands in this gel?

λ Figure 1d. The lane of GST-RPAP3-541-665 contains RUVBL2 band running noticeably lower than that in other lanes. Also the last lane contains bands around 100 kD that do not seem to belong to any component. Please clarify these issues. Also, please include GST only control for RUVBL2 pull-down in order to prevent non-specific interaction between RUVBL2 and the beads;

λ The first section ends by stating that "Instead of Pih1p acting as the central scaffold that connects the HSP90-recruitment factor Tah1p to the AAA+ ring, this role is taken by RPAP3...". This statement is not entirely correct because authors do not have data about PIH1D1 and how it may enhance or change the interactions.

λ Figure S1. Can authors eliminate the possibility that the density around the equatorial line of the RUVBL1/2 dodecamer is not RBD?

λ Figure S2. "IP" is not described. This figure is also cited as the evidence for TP disrupting R2 dodecamer but it does not really support this statement. In cryoEM data processing, authors did remove dodecamer classes. Are they sure that they are free of bound TP?

λ Figure S2e: what is the FSC in yellow? Please also provide orientation distribution;

λ It would be more satisfying to know that RBD-RUVBL2 interaction depends on salt concentrations due to the observed nature of interactions;

λ Figure S5a is missing and Figure S5f is mislabeled. Also, please orient the yeast and human R2TP densities similarly in order to compare the architecture of the TP binding of these two complexes;

λ Authors included biochemical characterization of HSP90 binding to R2TP that seem to suggest a tight interaction of HSP90. Perhaps some EM characterization of the HSP90-bound R2TP would help to clarify the location of the TPR domain;

λ Protein expression and purification is out of order. Perhaps the statement that "RUVBL1-RUBBL2 complexes used for cryo-EM studies were produced..." should be at the beginning of this section.

λ Although authors used co-expression experiments to assess interactions between RPAP3 and RUVBL2 and PIH1D1, the cloning section did not describe these constructs.

POINT-TO-POINT RESPONSE TO REVIEWERS

Reviewer #1

*In this manuscript Lorca and colleagues provide interesting data on the structure of the R2TP cochaperone and how human R2TP differs from its yeast counterpart. Whereas much progress has been made in recent years in elucidating the structure of *S. cerevisiae* R2TP, the human complex has remained poorly studied.*

We appreciate that the reviewer finds our work of interest, and we have addressed the concerns raised. We believe that the manuscript has improved after review, and we thank all the reviewers for their comments.

*While potentially interesting, this manuscript suffers from one major caveat: it acknowledges that RPAP3 is structurally different from yeast Tah1p, but falls short of applying that same logic to human R2TP as a whole by neglecting to consider (and even mention) any of the other subunits than the four that make up the yeast complex. Indeed, while there is much evidence showing that human R2TP also contains the prefoldin and prefoldin-like proteins PFDN2, PFDN6, URI1, UXT, PDRG1 (which is why it is usually referred to as R2TP/prefoldin-like or R2TP/PFDL) as well as RNA polymerase subunit POLR2E/RPB5 and WDR92/Monad, not one article published so far has conclusively demonstrated that human R2TP can exist solely as a combination of RUVBL1, RUVBL2, RPAP3 and PIH1D1 as is the case in *S. cerevisiae*.*

As indicated by the reviewer, the R2TP complex interacts with additional subunits, but possibly in the context of different client proteins or complexes. Many of the proteins the reviewer lists are associated with the involvement of R2TP with snoRNPs and RNA Pol II, but their connection to PIKKs or the MRN DNA repair complex for example, where R2TP is also involved, is less clear. It would not be completely accurate to present these as constitutive parts of the system. However, we take the reviewers general point, and have expanded the Introduction and Discussion to include and mention of the various proteins the reviewer cites.

Because all proteins used in this article were produced ectopically in bacteria and because the R2TP complex was spontaneously assembled by mixing together a subset of its subunits in vitro, the potential for artefactual interaction is high. The best procedure would have been to purify the complex directly from human cells and proceed with structural analysis. However, it is significantly more difficult to purify a fully-assembled protein complex from a cell lysate with enough purity to perform cryo-EM, which is likely why the authors opted for the recombinant protein shortcut. It is important that interpretation of the observations derived from this approach is tempered and presented as hypothesis rather than conclusions.

The reconstitution of complexes using proteins produced recombinant in *E. coli*, insect or mammalian cells is the standard and most widely used methodology in Structural Biology, that needs the production of the target complex in large amounts and high homogeneity.

Most crystal structures of protein complexes have been solved using recombinant proteins, and in most cases these structures reflect the structural and chemical nature of the complex as it happens in the cell, being the major limitation, compared with the purification of the endogenous complex, that some components of the “native” complex might be missing.

The major difficulties to purify the endogenous R2TP/PFLD complex will be the amount of material required for structural studies, and the heterogeneity in the subunit-occupancy of the complexes.

We have added some comments in the revised version (in the Discussion section) about the limitations of the approaches used.

Moreover, the C-terminal moiety of RPAP3, claimed here to be “novel”, has been reported previously and is already annotated as a bona fide protein domain (pfam13877) that is present in two other human proteins: CCDC103 and SPAG1. It is unfortunate that the manuscript failed to recognize this fact since interaction data for these two proteins tend to corroborate the hypothesis that this moiety is indeed involved in binding RUVBL2, as CCDC103 was shown to copurify with RUVBL2 (PMID: 25416956), while SPAG1 was observed with the RUVBL1-RUVBL2 interactor C12orf45 (PMID: 27173435).

We have fixed this in the revised version.

Lack of thorough review of the literature published so far on R2TP/PFDL and its cofactors can also be surmised by the authors claim that a number of biological aspects pertaining to the cochaperone are still poorly understood. For example, the manuscript states that “how RNA Pol II subunits are recruited to R2TP and how R2TP and HSP90 contributes to Pol II assembly is currently unknown” (line 62). While not entirely wrong, it is a rather bold statement as URI1, one of the R2TP/PFDL subunits disregarded by the authors, is a well-known direct interactor of RPB5 (a RNA Pol shared subunit) that was shown to affect assembly of all three nuclear RNA polymerases.

When we wrote the Introduction, we meant to indicate that the mechanistic and structural basis of Pol II recruitment are not well known. In comparison, the molecular basis of recruitment of clients by PIH1D1 is better understood.

Nonetheless, the reviewer is right that our Introduction was confusing about these concepts, and we have revised it accordingly.

The manuscript also states that “neither Pol II nor snoRNPs subunits contain [the PIH-binding] motif, and must therefore be recruited to R2TP through alternative mechanisms, yet to be described” (line 71). While PIH-binding motifs are indeed required for PIKK interaction through Tel2, binding to snoRNPs is mediated by NUFIP1 and ZnF-HIT domain proteins ZNHIT3 and ZNHIT6

Thank you for the comments. We have corrected Introduction accordingly.

. Additionally, interaction to the most recently discovered client complex of R2TP, U5 snRNP (regrettably not mentioned in this manuscript) appears to be mediated by a conjunction of both a PIH-binding motif protein (ECD) and a ZnF-HIT protein (ZNHIT2). Given that ZnF-HIT proteins were reported to interact with RUVBL2, it would be interesting for the authors to address that fact and raise the possibility of there being a competitive or collaborative interaction of RPAP3's C-terminal domain with this protein family.

Thank you for the comments. We have corrected the "Introduction" section accordingly. Also, the possibility of competition/collaboration of ZnF-HIT proteins and RPAP3 are now discussed in the Discussion section.

In sum, the present manuscript would benefit from 1) a serious effort to better review the literature,

We have addressed this issue.

2) the authors being more careful with claims of primacy in discovering RPAP3's C-terminal domain,

The reviewer is right and we have modified this.

3) addressing the fact that subunits are missing from the assembled complex and a more nuanced and conservative interpretation of the resulting structural data.

This is done. The Discussion section has been modified to address this issue.

This being said, the research presented in this manuscript is interesting enough to warrant publication in Nat Commun provided that 1) my comments above are taken into account in a revised manuscript, and 2) the manuscript is being reviewed by a protein structural biologist who might better look over the experimental methodology employed here (not being myself a specialist in structural studies). Understanding the structure of human R2TP is likely to accelerate research on this "most complex HSP90 cochaperone yet described".

Minor points:

1) Line 54-55: The first demonstration of an interaction with Pol II subunits comes from Jeronimo 2007. This same paper proposed the name RPAP3 for the first time (RNA Pol II Associated Protein 3). The reference Cloutier 2009 describes for the first time the 11-subunit R2TP/PFDL complex. Please clarify and add missing ref.

Corrected.

2) Line 168: I suggest continue using RUVBL1-RUVBL2-RBD all along the manuscript (instead of RBD-RUVBL1-RUVBL2).

Corrected.

Reviewer #2

RPAP3 provides a flexible scaffold for coupling HSP90 to the human R2TP co-chaperone complex. Fabrizio Martino et al.

The authors have done a series of experiments including cryo-EM and cross-linking mass spectrometry, which suggest that RPAP3 interacts within R2TP complex directly with ATPase domain of RUVBL2 through its RUVBL-binding-domain (RBD) located in the RPAP3 C-terminus. They also show that (in contrary to the small Tah1p molecule present in yeasts instead of RPAP3) RPAP3 spans to the opposite face of the RUVBL single ring and provides a platform for bringing HSP90 and R2TP client proteins together.

These novel findings uncover the architecture of R2TP complex and elucidate how the R2TP complex brings its substrate to the proximity of HSP90. Overall, the manuscript is well written and the data presented are of high quality.

We thank the reviewer for her/his comments. We believe that the manuscript has improved substantially after review, and we thank the reviewers for their comments.

I have only one comment - the authors use untagged RUVBL2 and RUVBL1 tagged with 3xmyc tag. However, the method part of the paper doesn't state whether the tag is C or N terminal and whether it could compromise RUVBL1 interaction with RPAP3. Have the authors tried to do a similar experiment to the one in Figure 1d with RUVBL1 untagged and RUVBL2 tagged with 3xmyc and would they be able to get the same results as shown in Figure 1D?

The 3xMyc tag is N-terminal in RUVBL1, and it was used to help distinguishing RUVBL1 and RUVBL2 in SDS-PAGE in the pull down experiments (otherwise they run in a very similar position).

For the cryo-EM experiments, the RUVBL1-RUVBL2 complex was purified using a N-terminal His-tag in RUVBL1 and non-tagged RUVBL2.

All this information is now clarified in the Methods section.

The N-terminus of RUVBL1 and RUVBL2 locate at the DII-domain face of the RuvBL ring, and therefore the N-terminal tags should not interfere with the binding to the RBD domain. We mention this now in the revised version.

As indicated by the reviewer, we have removed the tag in RUVBL1 and the protein does not bind RPAP3. This information can be found in the new Supplementary Figure 1.

Reviewer #3

(...) The novel aspect of this study is the discovery of the binding mode of the accessory RBD domain that provides additional anchor for the TP molecules on the ATPases.

We believe this finding is important to understand the architecture and function of human R2TP, and its comparison with yeast R2TP. We thank the reviewer for her/his comments, which have helped us to improve the manuscript significantly.

However, both structural and biochemical are preliminary and not sufficient to validate the proposed assembly model.

We think that we did not explain clearly enough important aspects of our results in the previous version, causing some misunderstandings that explain most of the issues raised by the reviewer. These problems have been solved in the revised version by improving the text and the figures.

In addition, we have performed new experiments and we add new data as response to the reviewer's concerns:

- New biochemical experiments and controls in Supplementary Figure 1.
- A new description/analysis of the disassembly of RUVBL1-RUVBL2 dodecamers by RPAP3-PIH1D1, together with the comparison of the effect of using RPAP3 versions missing the N-terminal TPR-containing region or containing only the RBD domain.
- A detailed analysis of RPAP3 stoichiometry has been included.
- The stoichiometry and shape of the RPAP3-PIH1D1 sub-complex has been analyzed by analytical centrifugation.
- We have generated a new RPAP3 construct, lacking the N-terminal region that contains the TPR domains. Cryo-EM images of an R2TP complex reconstituted using this truncated version map the TPRs in R2TP and help clarify the assignment of regions in the map.
- We have explained the modeling of the RBD better and we have focus the discussion on the secondary structure elements.
- We have improved the description of the flexible regions.
- We have improved the description of the image processing and modeling.
- We have put a lot of effort in trying to crystallize the RBD domain. We have just recently succeeded in obtaining crystals that diffracted to 1.8 Angstrom resolution, but they corresponded to a proteolytic fragment. Nonetheless, the structure corresponds to 1/3 of the domain, and supports the model.

Details of the new experiments and all changes in the new version of the manuscript can be found below, as response to each specific question.

Authors additionally characterized Hsp90 binding to R2TP biochemically but provided no structural follow-up.

We tested that R2TP was competent for HSP90 binding, as part of the characterization of the reconstituted complex.

Although we have worked on the binding of HSP90 to R2TP, we have been unable yet to characterize it structurally. Our preliminary data suggest that the R2TP-HSP90 was not stable in the conditions we used for cryo-EM and/or after vitrification, and we need to work out the stabilization of the complex. Also, the flexibility of the TPR regions in R2TP will probably be an important challenge when characterizing the interaction with HSP90.

In any case, we believe that the study of R2TP-HSP90 will require significant work that goes beyond the scope of this manuscript.

The major concerns are:

1) Authors seem to take it granted that human R2TP forms a hexamer. They stated that TP binding disrupted the R2 dodecamers but with minimal evidence. This is an important point, especially authors also state that RBD-RUVBL2 interaction is the “primary” interaction between R2 and TP. If RBD binds the ATPase domain of R2, it may well attach to the dodecamer without disruption of the oligomeric state. In a related concern 3), authors added ADP to the incubated R2TP, which is known to impact R2 oligomerization. Thus, it is not convincing at all that TP disrupts the dodecamer and the nature of human R2TP assembly remains unresolved;

Our results clearly show that R2 dodecamers are disrupted by RPAP3-PIH1D1 binding, but this information was not properly presented and described in the previous version. Part of this information was placed as supplemental information, and the tilted view of the dodecameric complex decorated by RBD domains, chosen for the display, does not help the reader to see that this is a RUVB1-RUVBL2 dodecamer decorated by three RBDs, and not a hexameric complex.

To clarify these concerns in the revised version:

- We have prepared a new section in Results dedicated to the effect of RPAP3-PIH1D1 and the RBD on the disassembly of RUVBL1-RUVBL2.
- We have prepared a new Figure 2, adding panels to address this issue and also coloring the structures to facilitate the analysis of the results.
- To further characterize the R2TP, we have reconstituted a R2TP complex using a truncated version of RPAP3 where the N-terminus and the TPR domains are removed (residues 395-665), but PIH1D1 binding is preserve. The complex was analyzed by cryo-EM and 2D averaging.

This new experiment provides three interesting pieces of information:

- RPAP3 residues 430-665 bind the RUVBL ring without affecting the dodecameric assembly of RUVBL1-RUVBL2 (Fig. 2A), but RPAP3 residues 395-665 bound to PIH1D1 disrupt the dodecamers.
- RPAP3 TPR-containing region can now be unambiguously assigned within the images of R2TP
- PIH1D1 localizes in the DII-domain side of the RUVBL ring.

Additional comments:

- Concerning the use of ADP: dodecameric RUVBL1-RUVBL2 complexes are formed in the presence of ADP (*Lopez-Perrote et al. NAR 2012*). The crystal structure of human RUVBL1 hexamers shows ADP trapped within the structure, despite ADP not being added during the purification (*Matias, P.M., Gorynia, S., Donner, P. and Carrondo, M.A. (2006) Crystal structure of the human AAA+ protein RuvBL1. J. Biol. Chem., 281, 38918–38929*).

Since the effects of ATP binding and/or hydrolysis on R2TP assembly are still uncharacterized, and ADP stabilizes the RUVBL1-RUVBL2 complex, we used ADP for the reconstitution experiments.

- To help a strict comparison, the images of RUVBL1-RUVBL2, RUVBL1-RUVBL2 bound to RBDs, and R2TP in this paper, were always obtained using the exact same RUVBL1-RUVBL2 sample, using the exact same buffer, and all containing ADP.

- It is worth mentioning that, in most of the RUVBL1-RUVBL2 complexes analyzed so far, hexamers are the functional assembly with most of the interactions with other proteins taken place at the DII-domain side of the ring. See for instance the recent structure of the INO80 complex (*Aramayo et al. NSMB, doi:10.1038/s41594-017-0003-7*). We include a comment on this in the revised version.

2) Despite cryoEM criteria suggest that the RBD-R2 structure is at a 3.6Å overall resolution, many problems are associated with this structural model.

The first is about the binding stoichiometry. Classes of cryoEM particles seem to show a distribution of 1, 2 or 3 RBD bound with R2.

We think that we did not explain these issues properly in the previous version. Most of the images (67 %) of the complex contain 3 RPAP molecules per RUVBL ring, and these particles were the ones used for processing.

To address this issue:

- We now show a classification analysis of the data, focused on identifying the number of RBDs, in the Supplemental Information. And we add a specific section in Results, dedicated to the characterization of the stoichiometry of the complex.

Complexes with just 1 or 2 RPAP3 molecules were a minority, likely reflecting that we did not saturate all the available RUVBL2 molecules under our experimental conditions, but that R2TP has the capacity to bind three RPAP3 per RUVBL ring.

- Be aware that RUVBL rings with 3 RBDs can generate projections with an appearance of 2 RBDs (like in the most abundant 2D averages), because 2 of the 3 RBDs coincide in the direction of the image projection in many of the possible orientations. Although this was mentioned in the previous version, we have tried to describe this better in this revised version, by preparing a new figure in Supplemental Information.

- The image processing strategy has been described better, and changes in the new Figure 4 (showing the structure with 3 RBDs and displaying only a slice of RUVBL1-RUVBL2-RBD for the high-resolution processing), should help the reader not to get confused about the stoichiometry of the complex (see later).

- The model cartoon in the final figure shows now 3 RBDs, in accordance with stoichiometry found in cryo-EM.

However, authors had to play a trick in order to increase signal-to-noise ratio by rotating all RBD into one RUVBL2 subunit such that the structure appears to be 1 RBD to R2 hexamer and this is what depicted in the final summary model (Fig. 7). Obviously, this is incorrect. Authors must make efforts in characterizing the stoichiometry by an independent method, as this property is very relevant to human R2TP structural model.

We are sorry that our processing strategy was not properly explained. The strategy was not meant to play tricks with the stoichiometry of the complex, but to make use of the best RBD data to improve the resolution.

Each R2TP has 3 RBDs bound to the RUVBL ring, but the quality and conformation of each RBD in the same molecule can be slightly different. Therefore, the use of a C3 can lower the resolution, but the use of no symmetry does not allow to average and classify all the available data.

We have followed a strategy designed by Sjors Scheres and described in Zhou et al, Genes and Development 2015

(<http://www.genesdev.org/cgi/doi/10.1101/gad.272278.115>), to address the issue of rotationally symmetric complexes when each element in the symmetry is slightly different in quality. In the Methods section of that article, the authors mention:

“In contrast to the central hub, the periphery of the Apaf-1 apoptosome, comprising the WD40 repeats and CytC, displays relatively low resolution. This feature may reflect the inherent property of the apoptosome: rigid conformation at the central hub and relatively mobile nature in the spokes.”

The details of their method were also described in that work:

“By applying a local mask around one spoke and performing 3D classification without any alignment, particles were classified by only considering the differences in the spoke region, other than the averaged differences from the entire apoptosome. The resulting class with the largest number of particles was chosen for 3D refinement. Only this spoke region and the linked central hub were aligned and refined locally, within 5°. This classification strategy proved to be useful for improving the local map quality. Because an apoptosome has a sevenfold symmetry, we rotated each particle around the symmetry axis seven times to put all seven spokes in the same position for local classification strategy. The original 134,919 particles that had been used to generate the 3.8 Å density map were treated as the first copy. These particles were rotated $360^\circ/7$ around the symmetry axis by adding $360/7$ to the column “_rlnAngleRot” in the Relion input star file, resulting in the second copy of the particles. The second copy of the

particles was similarly rotated by adding another 360/7 to the column “_rlnAngleRot,” generating the third copy of the particles. This operation was repeated four more times to generate the fourth, fifth, sixth, and seventh copies of the particles. For 134,919 particles, 944,433 spokes were rotated to the same position and classified with the local mask. The resulting largest class contained 196,815 spokes in the same configuration. The final reconstruction of these particles markedly improved the resolution in the spoke region to $\bullet 5 \text{ \AA}$, into which atomic coordinates from crystals structures were docked.”

We used this same strategy to improve the resolution for the structure of the RBD, which went from 3.81 Angstroms to 3.6 Angstroms, to obtain the best cryo-EM density for the RBD domain.

Since this seems to be confusing, we now show in Figure 4 a representation of the 3.81 Angstroms structure, containing 3 RBDs, and also the improved structure, within the figure, and the experimental details have been explained better in the manuscript. For the “improved structure”, we display a section of the RUVBL1-RUVBL2-RBD structure, containing a monomer of each component, to avoid misunderstandings about the stoichiometry of the complex.

Also, to avoid misunderstandings, the cartoon model in final figure also shows now 3 RBDs. We used only 1RBD initially for the cartoon to simplify the figure, but we are now aware that the fact that we find 3 RPAP3 molecules per R2TP should be very clear and consistent throughout the manuscript.

The second concern is about RBD structure itself. While it is clear that the region 541-665 binds R2, the exact interface between RBD and R2 is unambiguous. As authors maybe well aware of the limit of protein threading models, the actual registering of amino acids used to interpret the density of the helices may be incorrect. Authors did not provide any images of how side chains of RBD residues fit the density nor did they provide correlation coefficients for readers to judge the quality of the model.

To address this issue, we now describe in greater detail how the modeling was performed.

We believe that the register of the RBD structure is correct based on the quality of the RBD map (now shown in Supplementary Fig. 5, showing side chains for the RBD), the conservation of the helices in the prediction, the agreement between the prediction secondary structure elements and the cryo-EM map (now shown in a Supplemental movie), and the cross-links between RUVBL1 and RUVBL2 and the RBD which are consistent with the model.

But in addition, we have now crystallized a fragment of the RBD comprising most of helix 3, and all of helix 4 and helix 5. The structure of the fragment, now part of Figure 5, supports the modeling.

Side-chains visible for the RBD domain are shown in the supplemental figure. We also see some side chains in the interface between R2 and the RBD. Nonetheless, many side chains are not clear at this RBD/R2 interface. Thus, we have clarified in Results that the

interpretation of the RBD/R2 interface is mostly limited to the secondary structure elements, except when density for side chains are clear.

We have included the following information in the revised version to address this issue:

1. Densities for side chains are visible in many parts of the RBD map. We show now details of side chains of the RBD (Supplemental Figure 5).
2. We explain in the text the limitations in the interpretation of the RBD/R2 interface.
3. Supplemental Figure 4a shows the FSC between the cryo-EM and the RUVBL1-RUVBL2-RBD model, and Supplementary Table 1 includes a CC between the RUVBL1-RUVBL2-RBD model and the cryo-EM map provided by Phenix.
4. We now describe the methodology for modeling in greater detail in Methods.
5. We have prepared a movie as supplemental file, showing the fitting of I-tasser prediction to the cryo-EM map.
6. A control for the fitting of the homology model with the cryo-EM density was performed, and this is now described in the revised version. The fitting the RBD model in the reversed orientation is incompatible with the cryo-EM density and the crosslinks with RUVBL1 and RUVBL2
7. Finally, the new crystal structure of a fragment of the RBD is described.

We believe these changes clarify the results obtained to the reader, without affecting the key findings in the manuscript: the description of an alpha-helical domain at the C-terminal end of RPAP3 that binds RUVBL2 in the ATPase domain-side and directs R2TP assembly, the overall architecture of the R2TP core components, and the flexibility of the TPR regions at the opposite end of the RUVBL ring.

Biochemically, key residues at the interface should be disrupted and their effects on binding assessed.

Since we do not see many of the side chains in the RBD/R2 interface, we have decided to be more conservative in the interpretation of the RBD-R2 interface in the manuscript, and we specify this in the revised text.

3) Authors used ADP during reconstitution without providing any reasoning. What is the impact of ADP on R2TP assembly in their hands? Was this simply done to increase homogeneity? Which functional state of the ADP-bound human R2TP is?

As indicated above, ADP stabilizes RUVBL1-RUVBL2-containing complexes. Thus, ADP was used throughout the structural studies in this manuscript. We comment on this in the revised version.

To help a strict comparison, the images of RUVBL1-RUVBL2 and RUVBL1-RUVBL2 incubated with RBDs, also shown in this paper, were obtained in the same buffer used to obtain images of R2TP, also containing ADP.

These issues are now discussed and clarified in the revised version.

4) Structural characterization of TP binding to DII seems to be very difficult, which adds further confusion in stoichiometry. Authors presented two maps – one at 8.72 Å by classification and refinement of the entire asymmetric particles and another at 9.48 Å resulted from a masked region consists of DII and PIH1D1. The first map suggests that 3 RPAP3 are associated (Fig. 6b) whereas the second map suggests a single PIH1D1 is bound (Fig. 6a) to R2. On the other hand, biochemical data seem to suggest PIH1D1:RPAP3 is 1:1. How should these discrepancies be resolved? Curiously, there are no cross-links detected between PIH1D1 and any other components, which could result if PIH1D1 is primarily detached from DII. These issues seem to suggest problems either in sample preparation or structural characterization.

We agree with the reviewer that the structural characterization of the TP binding was confused in the previous version of the manuscript, which we have improved now, by including:

- A specific new section in the text describes that 3 RPAP3 molecules interact with each RUVBL1-RUVBL2 ring.
- The N-terminal region of RPAP3 is mapped within the end of the flexible regions in R2TP, after assembly and imaging of R2TP reconstituted with a truncated version of RPAP3 lacking the TPR regions and the most of the N-terminal part.
- New sedimentation velocity and sedimentation equilibrium experiments of the RPAP3-PIH1D1 have been performed and commented in Discussion.
- We have performed a new analysis of the data, selecting the particles with 3 RBDs, processed without symmetry and masking out the flexibly regions beyond the ring, aiming to define the PIH1D1 containing region in the 3-RBD R2TP complex.

This way, combined with other experiments in the manuscript, we determine the location of PIH1D1 in R2TP. We observe that the region is flexible, so resolution is limited. The density found for PIH1D1 can probably accommodate only 1 PIH1D1 molecule, as in yeast. But this is only a hypothesis at this stage and we discuss several alternatives in the revised version of the manuscript.

The new Figure 7 and the Discussion section have been modified to address these issues.

We have also decided to remove the fitting of the PIH1D1 homology models in the maps from this version, due to the limitations in resolution of the region.

The reviewer mentions the lack of cross-links detected between PIH1D1 and RUVBL1-RUVBL2. Given the location of PIH1D1 and the limited contacts with RUVBL1-RUVBL2 visible in the structure, it is conceivable that there are not many K residues in PIH1D1 sufficiently close to K residues in R2 to permit the crosslinking. Also, experimental conditions in the XL-MS experiments could need further tuning to detect cross-links affecting PIH1D1.

Other related concerns:

λ The titles in Supplementary Table II are very confusing. One refinement was assigned to R2-RBD-PIH1D1 and the other to R2TP subpopulation. If I understand it correctly,

it is the same data but refined differently. Please find a way to clarify these different reconstructions.;

The reviewer is right. The analysis of the flexible regions of R2TP was performed on the same dataset, but processed differently, as indicated in Results and Methods. We have now clarified this also in Table II in the revised version.

To address this point, we have also named each of the structures solved and deposited in the EMD-base, in the text and Tables.

λ Some example density maps are provided in Fig. S2 but they appear to be the well conserved regions within R2 rather than regions within RBD. It would be important to validate the tracing of RBD;

The reviewer is correct. We know show, in addition, some well defined regions in the model in the new Supplementary Fig. 5.

λ Figure 1b. All constructs shown in this figure resulted in solubilization of PIHD1D1. Did authors have examples of constructs that did not solubilize PIHD1 that can provide a support for the proposed role of 400-420 (for instance, GST only)?

This experiment is shown in Fig. 1c. Whereas GST-RPAP3 (267-420) binds the CS domain of PIHD1 (Fig. 1b), GST-RPAP3 (267-400) does not. We have repeated this experiments many times with different versions of RPAP3.

What are the upper bands in this gel?

Some of the experiments show some contaminants, and this is now indicated in the legend of the figure.

λ Figure 1d. The lane of GST-RPAP3-541-665 contains RUVBL2 band running noticeably lower than that in other lanes. Also the last lane contains bands around 100 kD that do not seem to belong to any component. Please clarify these issues.

The same gel contains pull down experiments using full length RUVBL2 and RUVBL2 where the DII domains were truncated, which runs lower. Although this was indicated in the top of the gel, it was not fully clear, and we have modified the top labels in the figure to make this clearer.

Also, please include GST only control for RUVBL2 pull-down in order to prevent non-specific interaction between RUVBL2 and the beads;

The control using GST only is now shown in Supplemental Figure 1.

λ The first section ends by stating that “Instead of Pih1p acting as the central scaffold that connects the HSP90-recruitment factor Tah1p to the AAA+ ring, this role is taken by RPAP3...”. This statement is not entirely correct because authors do not have data about PIHD1 and how it may enhance or change the interactions.

The reviewer is completely correct, and we have rephrased this sentence as “RPAP3 takes at least part of this role”.

λ Figure S1. Can authors eliminate the possibility that the density around the equatorial line of the RUVBL1/2 dodecamer is not RBD?

The tilted view of the RUVBL1-RUVBL2-RBD structure shown in Figure S1 was confusing. The new panels in Figure 2 should help to clarify this issue. The comparison between the images and 3D structures of RUVBL1-RUVBL2 (this work and our previous work in Lopez-Perrote et al., NAR 2012) and RUVBL1-RUVBL2-RBD showed a domain decorating the ATPase face side of the RUVBL ring. The cross-links between residues at the RBD and the C-terminus of RUVBL1 and RUVBL2 can only be explained by the RBD binding to the ATPase side.

Also, the biochemistry (Figure 1) showed that removing the DII domains from RUVBL2 did not affect the interaction with the RBD.

We think that the new section in Results and the new Figure 2 addresses this issue.

λ Figure S2. “IP” is not described. This figure is also cited as the evidence for TP disrupting R2 dodecamer but it does not really support this statement.

This panel is part of the new Figure 3 and we have changed the label.

In cryoEM data processing, authors did remove dodecamer classes. Are they sure that they are free of bound TP?

Yes, we are sure.

Particles of the cryo-EM experiment classified as RUVBL1-RUVBL2 dodecamers were processed independently, and they were identical to the images and the structure of the RUVBL1-RUVBL2 dodecamers we have solved before (Lopez-Perrote et al, NAR 2012).

λ Figure S2e: what is the FSC in yellow?

This was an error in the color of the label/legend, which has been corrected.

Please also provide orientation distribution;

Done, and shown in a supplemental figure.

λ It would be more satisfying to know that RBD-RUVBL2 interaction depends on salt concentrations due to the observed nature of interactions;

Although differences in charges between the surface of the ATPase domain face in RUVBL1 and RUVBL2 are notable, we have removed this panel from the final version of the manuscript. Increasing salt concentration to 300 mM did not abolish the interaction between the GST-RPAP3 and RUVBL2, and we didn't see any visible effect in the quantity of the pulled down RUVBL2 with GST-RPAP3.

λ Figure S5a is missing and Figure S5f is mislabeled. Also, please orient the yeast and human R2TP densities similarly in order to compare the architecture of the TP binding of these two complexes;

Supplemental Fig. 5 has changed. Yeast R2TP is now in the new Figure 7, and we orient the two complexes showing a similar view.

λ Authors included biochemical characterization of HSP90 binding to R2TP that seem to suggest a tight interaction of HSP90. Perhaps some EM characterization of the HSP90-bound R2TP would help to clarify the location of the TPR domain;

Although we have worked on the binding of HSP90 to R2TP, we have been unable yet to characterize it structurally. Our preliminary data suggest that the R2TP-HSP90 was not stable in the conditions we used for cryo-EM and/or after vitrification, and we need to work out the stabilization of the complex.

In the revised version, removal of the TPR region in Figure 2 now helps to assign these domains to the flexible regions in the R2TP complex, although they cannot be solved at high resolution due to their flexibility.

λ Protein expression and purification is out of order. Perhaps the statement that “RUVBL1-RUBBL2 complexes used for cryo-EM studies were produced...” should be at the beginning of this section.

We used a slightly different version of RUVBL1-RUVBL2 for pull down experiments and cryo-EM. A N-terminal Myc-tagged version of RUVBL1 was used for the pull downs, to help distinguishing RUVBL1 and RUVBL2 in SDS-PAGE. For cryo-EM, we used the methods described in Lopez-Perrote et al (NAR 2012) to produce highly pure and concentrated protein suitable for structural studies.

We have clarified this in the Methods section.

λ Although authors used co-expression experiments to assess interactions between RPAP3 and RUVBL2 and PIH1D1, the cloning section did not describe these constructs.

We have clarified this in the Methods section.

REVIEWERS' COMMENTS:

Reviewer #1 (Remarks to the Author):

In this revised manuscript, Martino et al addressed to my satisfaction the comments/criticisms raised in my previous review. I recommend to accept as is.

Reviewer #2 (Remarks to the Author):

The authors have addressed all of my comments and concerns and I think the manuscript has much improved since its first submitted version. I would therefore recommend acceptance of the manuscript for publication in Nature Communications.

Reviewer #3 (Remarks to the Author):

Martino et al. revised the previously submitted work on human R2TP structural studies. Authors provided several pieces of additional data that indeed enhanced the model at hand. In particular, authors added careful examination of stoichiometry and revealed a new paradigm that human R2TP may have a varying T:P ratio when bound with the two ATPases. I am fine with not addressing why in this manuscript but to show that this is the case. Additionally, a new crystal structure of a partial RBD domain of RPAP3 was obtained that supports the tracing of the RBD model. Overall, I am satisfied with the conclusions made by the authors except for the following minor criticisms:

1. The references throughout the revised manuscript are completely messed up. They are not citing what they are supposed to. Please correct them.
2. Authors under-utilized the crystal structure of fragment 578-624. Instead of making use of this structure that comprises most of the modeled RBD (541-665), authors used a computational model that is obviously inferior to the crystal structure. I suggest authors substitute 578-624 in the modeled 541-665 fragment with the crystal structure. This will only leave two very small sections modeled;
3. There is no description of how the crystal structure was solved (Molecular Replacement by what search model or SAD by what heavy atoms?);
4. Page 8, end of second paragraph. Authors emphasize, based on the identified RPAP3-RUVBL2 interaction, that the primary interaction between TP and RUVBL1/2 is through RPAP3. This is inaccurate because they have not eliminated a role of PIH1D1 in this interaction;
5. The section titled "RPAP3-PIH1D1 but not RBD disrupts dodecameric RUVBL1-RUVBL2". Why didn't authors use the full-length of RPAP3 to really nail down the question if RPAP3 is sufficient (without PIH1D1) to disrupt the dodecamer?
6. Did the sedimentation equilibrium analysis provide an equilibrium constant between P and T? What is it?
7. Figure 8 caption. The model of human R2TP where three RPAP3 are depicted can be misleading. As authors describe in the Discussion that a functional stoichiometry is not known, this model should be revised to reflect this important point. It may well be that one PT is associated with the functional R2TP.

POINT-TO-POINT RESPONSE TO REVIEWERS

Reviewer #1 (Remarks to the Author):

In this revised manuscript, Martino et al addressed to my satisfaction the comments/criticisms raised in my previous review. I recommend to accept as is.

Thanks

Reviewer #2 (Remarks to the Author):

The authors have addressed all of my comments and concerns and I think the manuscript has much improved since its first submitted version. I would therefore recommend acceptance of the manuscript for publication in Nature Communications.

Thanks

Reviewer #3 (Remarks to the Author):

Martino et al. revised the previously submitted work on human R2TP structural studies. Authors provided several pieces of additional data that indeed enhanced the model at hand. In particular, authors added careful examination of stoichiometry and revealed a new paradigm that human R2TP may have a varying T:P ratio when bound with the two ATPases. I am fine with not addressing why in this manuscript but to show that this is the case. Additionally, a new crystal structure of a partial RBD domain of RPAP3 was obtained that supports the tracing of the RBD model. Overall, I am satisfied with the conclusions made by the authors except for the following minor criticisms:

Thanks

1. The references throughout the revised manuscript are completely messed up. They are not citing what they are supposed to. Please correct them.

An error took place when the final revised manuscript file was compared with the first submission document, to indicate the changes. Track changes were required for the resubmission to the journal. Unexpectedly, the document with track changes retained the old, rather than the new, list for references, and I did not notice the error.

This has been fixed, but I have also carefully checked that references are now correct.

2. Authors under-utilized the crystal structure of fragment 578-624. Instead of making use of this structure that comprises most of the modeled RBD (541-665), authors used a computational model that is obviously inferior to the crystal structure. I suggest authors substitute 578-624 in the modeled 541-665 fragment with the crystal structure. This will only leave two very small sections modeled;

Thanks for the suggestion, but we have reasons to prefer leaving it as it is in the current version.

We did not use the crystal directly, because only helices 4 and 5 were complete. Helix 3 in the crystal was partial. Helix 6 in the crystal pointed to a different direction to that in the cryo-EM.

The conformation of such a short segment will be strongly influenced by crystal packing, so the overall conformation observed in the cryoEM is more authoritative. The crystal structure does however give an unambiguous high-resolution view of the side chain threading for that segment of the domain, and this information was used for the modeling.

We have added one sentence in the appropriate section (page 13 of the revised version), to clarify this point.

3. There is no description of how the crystal structure was solved (Molecular Replacement by what search model or SAD by what heavy atoms?);

We solved the crystal structure with ARCIMBOLDO software, which find the secondary structures only if the dataset have high resolution. Then structure was auto built with Buccaneer and refined with Refmac.

The corresponding Methods section has been clarified.

4. Page 8, end of second paragraph. Authors emphasize, based on the identified RPAP3-RUVBL2 interaction, that the primary interaction between TP and RUVBL1/2 is through RPAP3. This is inaccurate because they have not eliminated a role of PIH1D1 in this interaction;

The reviewer is right that this interpretation cannot be derived solely from the results shown in Figure 1.

Instead, the experiment in Supplemental Fig. 1a shows that a TP construct where the RBD domain was removed does not bind RuvBL2. Thus, the RBD domain in RPAP3 is the only domain essential to maintain the RPAP3-RUVBL2 interaction.

This was not well explained in the main text, and a reference to Supplemental Fig. 1a was not mentioned, and we have now fixed it.

5. The section titled "RPAP3-PIH1D1 but not RBD disrupts dodecameric RUVBL1-RUVBL2". Why didn't authors use the full-length of RPAP3 to really nail down the question if RPAP3 is sufficient (without PIH1D1) to disrupt the dodecamer?

Thank you for the idea. We did not test this, because we were mostly focused on the effects of RPAP3-PIH1D1 on RUVBL1-RUVBL2 as part of the assembly of R2TP. Also, the comparison between RPAP3 residues 430-665 which bind the RUVBL ring without affecting the dodecameric assembly of RUVBL1-RUVBL2, and RPAP3 residues 395-665

bound to PIH1D1, which does disrupt the dodecamer, strongly suggest an effect of PIH1D1.

To address this issue, we have now incubated RUVBL1-RUVBL2 with full length RPAP3 (without PIH1D1), and the complex has been analyzed in a FEI Talos Arctica equipped with a Falcon III detector.

When the images were processed, the great majority corresponded to end-views of the complex. RUVBL1-RUVBL2 dodecamers were not found. This can be interpreted as an indication that the RUVBL1-RUVBL2 dodecamers had disassembled, since hexameric RUVBL1-RUVBL2 has a much higher propensity to top-views than side-views. Having said this, top-views are similar in either hexamers or dodecamers, and thus, we cannot conclude without ambiguity. The SDS-PAGE and a representative top-view of the experiment are shown here to the reviewer.

Excess RPAP3 incubated with RUVBL1-RUVBL2 (different amounts loaded in the gel)

2D averages. The difference in intensity within the ring would be compatible with RPAP3 binding to RUVBL1-RUVBL2 in alternating subunits of the ring.

In addition, other less abundant views were observed in the micrographs, but we were unable to interpret them as either hexamers or dodecamers, and averages were significantly blurry. Maybe RPAP3 is even more flexible when bound to RUVBL1-RUVBL2 without the attachment provided by PIH1D1.

Thus, these experiments need some re-evaluation and we will follow this idea in future works.

To address this issue in the revised manuscript, we have added this sentence in page 10: *"These experiments do not discard that RPAP3 alone could be sufficient to disrupt the RUVBL1-RUVBL2 dodecamer."*

6. Did the sedimentation equilibrium analysis provide an equilibrium constant between P and T? What is it?

Unfortunately, the experimental set up was designed to estimate the molecular weight and the stoichiometry of the TP complex. To obtain equilibrium constants we should have designed a different experimental setup, exploring several concentrations and T/P ratios. This was not done, as our goal was to define the stoichiometry of the RPAP3-PIH1D1 complex.

7. Figure 8 caption. The model of human R2TP where three RPAP3 are depicted can be misleading. As authors describe in the Discussion that a functional stoichiometry is not known, this model should be revised to reflect this important point. It may well be that one PT is associated with the functional R2TP.

The reviewer is right.

We have modified the figure to reflect that the number of RPAP3 per RUVBL ring in vivo is not known. This change in the figure is reflected in some additional sentences in the Discussion section.